# LONGVIDEOBENCH: A Benchmark for Long-context Interleaved Video-Language Understanding

**Haoning Wu    Dongxu Li    Bei Chen    Junnan Li**[*]

Rhymes AI

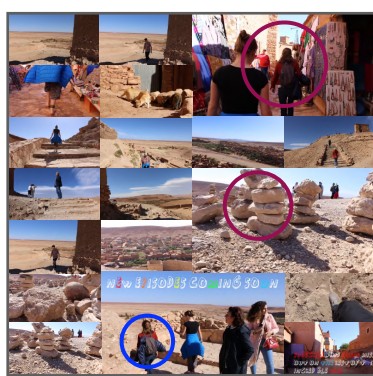
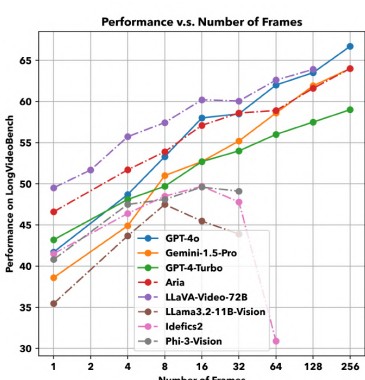

Figure 1: **(Left)** LONGVIDEOBENCH features *referring reasoning* questions, with a *referring query* that references particular video contexts (*i.e. referred context*) to answer questions about. **(Right)** For top-tier models, LONGVIDEOBENCH requires more input frames to obtain better performance.

## Abstract

Large multimodal models (LMMs) are processing increasingly longer and richer inputs. Albeit the progress, few public benchmark is available to measure such development. To mitigate this gap, we introduce LONGVIDEOBENCH, a question-answering benchmark that features video-language interleaved inputs up to an hour long. Our benchmark includes 3,763 varying-length web-collected videos with their subtitles across diverse themes, designed to comprehensively evaluate LMMs on long-term multimodal understanding. To achieve this, we interpret the primary challenge as to accurately *retrieve* and *reason over* detailed multimodal information from long inputs. As such, we formulate a novel video question-answering task termed *referring reasoning*. Specifically, as part of the question, it contains a *referring query* that references related video contexts, called *referred context*. The model is then required to reason over relevant video details from the referred context. Following the paradigm of referring reasoning, we curate 6,678 human-annotated multiple-choice questions in 17 fine-grained categories, establishing one of the most comprehensive benchmarks for long-form video understanding. Evaluations suggest that the LONGVIDEOBENCH presents significant challenges even for the most advanced proprietary models (*e.g.* GPT-4o, Gemini-1.5-Pro, GPT-4-Turbo), while their open-source counterparts show an even larger performance gap. In addition, our results indicate that model performance on the benchmark improves only when they are capable of processing more frames, positioning LONGVIDEOBENCH as a valuable benchmark for evaluating future-generation long-context LMMs.

---

[*]Corresponding Author. Dataset and leaderboard at: `https://longvideobench.github.io`

38th Conference on Neural Information Processing Systems (NeurIPS 2024) Track on Datasets and Benchmarks.

# 1 Introduction

Recent foundation models are processing inputs of longer contexts, with a growth from 2K tokens as in LLaMA [Touvron et al., 2023], to 128K as in GPT-4 [OpenAI, 2024a] and further into millions in models like Gemini-1.5-Pro [Team, 2024]. To measure such development, most benchmarks focus on text-only inputs [Hsieh et al., 2024, Wang et al., 2024a, gkamradt, 2024], while those for long multimodal context remain lacking. In this regard, the task of understanding long-duration videos, such as those extending up to hours, is considered a promising testbed. However, existing video benchmarks exhibit strong single-frame bias. Namely, their results do not improve even models can process more frames. This longstanding issue has continued to be a pain in the neck for video understanding, making evaluation of long-context multimodal inputs a significant challenge.

To address this challenge, this work introduces LONGVIDEOBENCH, a video understanding benchmark that measures the progress of LMMs in processing hour-long subtitled videos. In contrary to findings from previous benchmarks, we observe consistent performance improvements when an LMM is capable of processing a larger number of frames (Fig. 1 (b)). To achieve this, we begin by identifying two capabilities essential for long-context multimodal understanding. First, akin to the needle in a haystack (NIAH) evaluation for text LLMs [gkamradt, 2024], effective LMMs must be adept at perceiving specific multimodal details in response to user queries, a task that becomes harder with longer input lengths. Second, in addition to recalling specified elements, the model must be able to relate them and reason about them coherently and contextually. This challenges the model to interpret and integrate large volumes of multimodal information meaningfully.

To effectively evaluate these abilities, we design ***referring reasoning*** (Fig. 1 (a)) as the foundation task for our benchmark. In particular, this task initially introduces a *referring query*. It references particular video contexts, which are termed the *referred context*. Subsequently, the model is presented with a question related to this referred context. This question tests the model's multimodal understanding capabilities, such as visual perception and relational reasoning. To achieve good performance in referring reasoning, models have to interpret the referring query and accurately recall the referred context from the long-context inputs. In addition, they need to perform complex multimodal reasoning. These challenges are closely aligned with the required capabilities as outlined previously.

Following the task of referring reasoning, the LONGVIDEOBENCH contains 6,678 multiple-choice questions on 3,763 videos. These videos are diverse in their themes, including movies, news, life and knowledge, covering 4 progressive duration groups: *8-15 seconds*, *15-60 seconds*, *3-10 minutes*, and *15-60 minutes*, making LONGVIDEOBENCH widely relevant for real-world video applications. Videos are also accompanied with original or transcribed subtitles, which challenges the model to understand long-context interleaved multimodal inputs.

We incorporate *perception* and *relation* questions in the benchmark. Specifically, perception questions require the model to perceive visually on an individual referred video scene, such as to recognize objects, attributes and events. In contrast, relation questions require the model to associate multiple scenes within the referred context, and answer questions about their temporal ordering, attribute change or to track referred objects. These questions are further divided into 17 fine-grained categories, with human-annotated choices, covering a wide range of video understanding tasks.

Our contributions are summarized in three-fold:

1. We introduce LONGVIDEOBENCH (Tab. 1), a multi-choice question-answering benchmark for long-context multimodal video understanding. Our benchmark consists of 6,678 human-crafted comprehensive questions posed on vary-length videos up to an hour long on diverse themes, widely relevant for video understanding applications in the wild.

2. We propose the task of *referring reasoning* to effectively address the longstanding issue of single frame bias in video understanding metrics. As a result, models have to be capable of processing effectively more frames, longer multimodal inputs to improve performance. This requirement distinguishes LONGVIDEOBENCH from existing video benchmarks;

3. We evaluate comprehensively the proprietary and open-source models to understand their long-context multimodal modeling capabilities. Our results demonstrate significant challenges posed by LONGVIDEOBENCH. In addition, the evaluation results show intriguing insights into deficiencies of existing models, thereby offering valuable directions for future research on multimodal long-context understanding.

Table 1: The LONGVIDEOBENCH and popular benchmarks for video LMMs. The [(HT)] denotes the benchmarks split test sets with hidden answers to avoid contamination.

| Benchmark | Labels | #Eval Videos | #Eval QAs | Avg Duration (s) | Theme Category | Interleaved? |
|---|---|---|---|---|---|---|
| MSVD-QA [Xu et al., 2017] | Auto | 520 | 13,157 | 10 | Everyday Life | ✗ |
| MSRVTT-QA [Xu et al., 2017] | Auto | 2,990 | 72,821 | 15 | Everyday Life | ✗ |
| ActivityNet-QA Yu et al. [2019] | Human | 800 | 8,000 | 180 | Everyday Life | ✗ |
| NeXT-QA [Xiao et al., 2021] | Human | 1,000 | 8,564 | 44 | Everyday Life | ✗ |
| MVBench [Wang et al., 2023] | Auto | 4,000 | 4,000 | 16 | Life, Human Action, Movie | ✗ |
| EgoSchema [Mangalam et al., 2023] | Auto | 5,031 | 5,031[(HT)] | 180 | Life, Human Action | ✗ |
| MovieChat-1K [Song et al., 2023] | Human | 130 | 1,950 | 500 | Movie | ✗ |
| **LONGVIDEOBENCH (ours)** | Human | 3,763 | 6,678[(HT)] | 473 | Life, Movie, Knowledge, News | ✓ |

Table 2: Definition of 17 categories of *referring reasoning* questions in the LONGVIDEOBENCH.

| Level | Task | Type of *referring query* (Q) | Type of Target Answer | Code | # |
|---|---|---|---|---|---|
| **Perception** (L1, 3204) | SCENE-REFERRED EVENT | a scene | an event that happens in Q | S2E | 410 |
| | SCENE-REFERRED OBJECT EXISTENCE | a scene | an object that exists in Q | S2O | 403 |
| | SCENE-REFERRED OBJECT ATTRIBUTE | a scene$^{q_1}$+an object$^{q_2}$ | an attribute of $q_2$ in $q_1$ | S2A | 403 |
| | EVENT-REFERRED OBJECT | an event | an object that participates Q | E2O | 393 |
| | OBJECT-REFERRED EVENT | an object | an event while Q appears | O2E | 401 |
| | TEXT-REFERRED EVENT | a subtitle | an event concurrent with Q | T2E | 398 |
| | TEXT-REFERRED OBJECT EXISTENCE | a subtitle | an object that exists while Q | T2O | 387 |
| | TEXT-REFERRED OBJECT ATTRIBUTE | a subtitle$^{q_1}$+an object$^{q_2}$ | an attribute of $q_2$ while $q_1$ | T2A | 402 |
| **Relation** (L2, 3474) | EVENT BEFORE/AFTER EVENT | an event | an event that happens before/after Q | E3E | 406 |
| | OBJECT BEFORE/AFTER OBJECT | an object | an object that appears before/after Q | O3O | 394 |
| | SEQUENCE OF SCENES | multiple scenes | the sequential order among Q | SSS | 398 |
| | SCENE-REFERRED OBJECT TRACKING | a scene$^{q_1}$+an object$^{q_2}$ | another scene that $q_2$ appears | SOS | 381 |
| | SCENE-REFERRED OBJECT ATTRIBUTE CHANGE | two scenes$^{q_1,q_2}$+an object$^{q_3}$ | attribute change of $q_3$ from $q_1$ to $q_2$ | SAA | 375 |
| | EVENT BEFORE/AFTER TEXT | a subtitle | an event that happens before/after Q | T3E | 401 |
| | OBJECT BEFORE/AFTER TEXT | a subtitle | an object that appears before/after Q | T3O | 391 |
| | TEXT-REFERRED OBJECT TRACKING | a scene$^{q_1}$, an object$^{q_2}$ | subtitle at $q_2$'s appearance other than $q_1$ | TOS | 380 |
| | TEXT-REFERRED OBJECT ATTRIBUTE CHANGE | two subtitles$^{q_1,q_2}$+an object$^{q_3}$ | attribute change of $q_3$ from $q_1$ to $q_2$ | TAA | 348 |

## 2 The *Referring Reasoning* Task

In this section, we first identify the primary challenges for multimodal long-context understanding. To reflect these challenges, we further define *referring reasoning*, the foundational task for LONGVIDEOBENCH. We introduce its general task scheme and specific categories as follows.

**Challenges for the LONGVIDEOBENCH.** Similar to challenges identified in text-only long-context benchmarks [gkamradt, 2024, Hsieh et al., 2024], the LONGVIDEOBENCH designs question-answering tasks to reflect the following two major difficulties in understanding long videos:

First, **retrieving details** from long videos. Existing studies [gkamradt, 2024, Team, 2024] notice that LLMs or LMMs often struggle to extract specific details from long sequences. To accurately assess this capability in the domain of long videos, the tasks in LONGVIDEOBENCH demand a focus on granular details such as *objects, events*, or *attributes*, rather than a summary or topic overview.

Second, **reasoning contextual relations** in long videos. According to Hsieh et al. [2024], beyond mere retrieval, it is significantly challenging for models to reason about the relationships among extensive inputs. Questions in LONGVIDEOBENCH are therefore designed to compel LMMs to analyze the interconnections among diverse content within a long video to derive the correct answer.

**General Scheme for *Referring Reasoning*.** To effectively measure model performance against aforementioned challenges, we establish the *referring reasoning* task as the fundamental paradigm for LONGVIDEOBENCH. Each question begins by describing a *referring query*, pinpointing one or multiple moments from the video. These video moments, composed of frames and subtitles, are denoted as *referred context*. A specific question body follows the referring query, which requires the model to reason over the referred context to deduct the answer. We employ the multiple-choice question format, where several distracting options are provided alongside the correct answer option.

**Two Levels: Perception *and* Relation.** We divide *referring reasoning* questions into two levels. In **(L1) Perception**, the referring query references a single moment of the video. Then, a question body is posed to ask about the visual perception of a specific concept in the referred moment, such as object, action, or event. (L1) questions mainly challenge models on locating the referred context from the long inputs and understand its visual information. In **(L2) Relation**, the referred context spans across multiple moments of the video. These moments are either related with a specific sequential

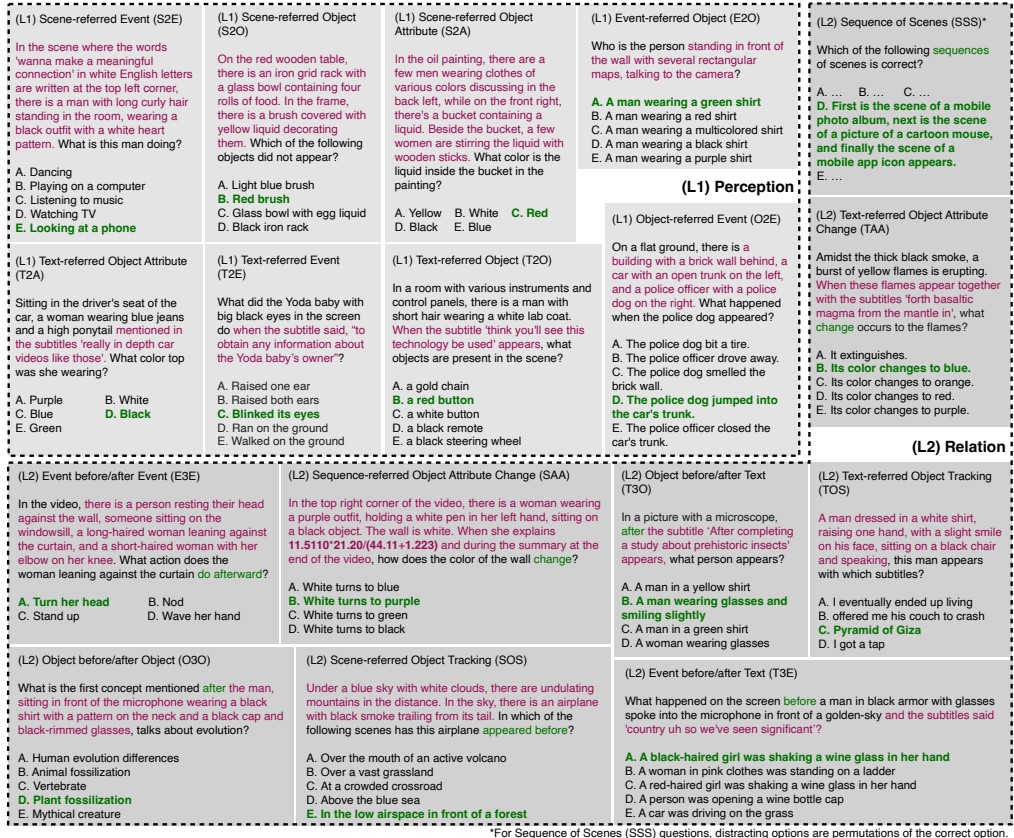

Figure 2: Examples of 17 categories of *referring reasoning* questions in the LONGVIDEOBENCH.

order (before/after/concurrent) or containing the same concept (*e.g.* the same object appears in these moments). The question is then posed regarding the relations of the moments, and answering these questions require models to not only locate the referred moments, but further reason over their relations. This makes (L2) questions in general more challenging than (L1) questions.

**17 Finer-grained Question Categories.** We further subdivide the two levels of questions into 17 finer-grained categories, dividing based on the type of referring query and the type of target answer. As listed in Tab. 2, given interleaved multimodal inputs, the referring query could either be describing a scene, an event, or an object from the video frames, or be narrating a sentence or a phrase from the text subtitles. The target answer typically is about a visual concept (an event, object, or attribute) from one of the referred moments, with two exceptions: the SEQUENCE OF SCENES (SSS) category requires to answer the correct sequential order of multiple ($> 3$) scenes in the video, and the TEXT-REFERRED OBJECT TRACKING (TOS) requires to answer the specific subtitle while a given object appears.

## 3 Dataset Construction

In this section, we discuss the dataset construction for the LONGVIDEOBENCH. We first define the category and duration groups of videos (Sec. 3.1), then we introduce the process of collecting and creating interleaved video-subtitle data (Sec. 3.2), lastly we elaborate on the human annotation process to collect high-quality referring questions and answers for LONGVIDEOBENCH (Sec. 3.3).

### 3.1 Groups of Videos

**Progressive Duration Groups.** In LONGVIDEOBENCH, we aim to not only evaluate LMMs on ultra-long videos, but analyze how their ability changes from short videos (*about 10s*) to long (*hour-long*). In light of this, we propose to collect videos in four progressive duration groups, as listed in Tab. 3.1. The first two groups contain shorter videos of length *(8s, 15s]* and *(15s, 60s]*, whereas the latter two duration groups contain long videos of length *(180s, 600s]* and *(900s, 3600s]*. The four groups not

Table 3: Statistics of videos in LONGVIDEOBENCH, by duration groups and video layouts.

| Duration Group | (8s, 15s] | | | | (15s, 60s] | | | | (180s, 600s] | | (900s, 3600s] | |
|---|---|---|---|---|---|---|---|---|---|---|---|---|
| Source Platform | Landscape | | Portrait | | Landscape | | Portrait | | Landscape | | Landscape | |
| Statistics | Duration | #Videos | Duration | #Videos | Duration | #Videos | Duration | #Videos | Duration | #Videos | Duration | #Videos |
| | 11.06 | 546 | 11.93 | 338 | 33.88 | 551 | 38.59 | 374 | 389 | 986 | 1408 | 966 |

Table 4: Statistics of videos in LONGVIDEOBENCH, by category groups (*in two-letter codes, as defined in Sec. 3.1*) and video layouts (*LS*: Landscape; *PT*: Portrait.)

| Category Group | Movie Recaps | | Everyday Life | | | | | | News Program | Knowledge | | | | |
|---|---|---|---|---|---|---|---|---|---|---|---|---|---|---|
| | (MR) | | LT | | LV | | LC | | (NP) | KA | KH | KG | KS | KC |
| Source Platform | LS | PT | LS | PT | LS | PT | LS | PT | LS | LS | LS | LS | LS | LS |
| #Channels | 18 | 4 | 10 | 6 | 7 | 5 | 8 | 5 | 12 | 5 | 8 | 7 | 10 | 7 |
| #Downloaded Videos | 7679 | 1106 | 2230 | 2532 | 1009 | 1891 | 2731 | 4706 | 24002 | 2010 | 2900 | 1280 | 1350 | 335 |
| #Annotated Videos | 352 | 160 | 343 | 173 | 338 | 179 | 336 | 203 | 329 | 327 | 336 | 330 | 200 | 160 |

only cover the duration ranges of existing video understanding benchmarks, but also provide a unique hour-long subset to further expand the video length beyond existing benchmarks.

**Category Groups.** Existing LMM benchmarks for long videos typically focus on a specific category of videos, *e.g.* egocentric videos [Mangalam et al., 2023], or movies [Song et al., 2023, Zhang et al., 2023a]. In comparison, LONGVIDEOBENCH is a more comprehensive benchmark that covers diverse categories of contents. The videos in LONGVIDEOBENCH are collected from 99 different channels for landscape videos and 20 channels for portrait videos, in the 10 following categories: Movie Recaps (*MR*); three life-related categories: Travel Guides (*LT*), Life Vlogs (*LV*), Cooking/Recipes (*LC*); News Programs (*NP*); and five knowledge-related categories: Art (*KA*), History (*KH*), Geography (*KG*), STEM (*KS*), Computer Science (*KC*). As listed in Tab. 4, LONGVIDEOBENCH includes a sufficient number of videos from all 10 category groups, spanning over a diverse distribution.

## 3.2 Video and Subtitle Collection

The video and subtitle collection process is illustrated in Fig. 3. First, all videos with at least 720P resolution from the 119 channels are downloaded. After downloading the videos, for the source platforms that provide transcribed subtitles, we remove the videos without transcribed subtitles or with non-English subtitles. For those videos without provided transcribed subtitles, we employ Whisper-V3-Large [OpenAI, 2024b] to generate subtitles for them. These videos are further sampled to cover different topics uniformly. Finally, we evaluate their video quality via Q-Align [Wu et al., 2024] and remove especially low-quality videos to ensure that all videos have scores $> 0.25$ (in range $[0, 1]$). Remaining videos are further manually filtered by annotators (in Sec. 3.3) to the final 3,763 videos.

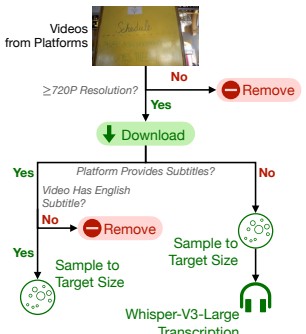

Figure 3: Video collection for LONGVIDEOBENCH, ensuring all videos have subtitles.

Subtitles are important for multimodal video understanding, as they provide vital text information from human speech and reduce ambiguity from pure visual scenes. Aligning with the way humans watch videos with subtitles, in LONGVIDEOBENCH, we require LMMs to receive the text subtitles simultaneously with concurrent frames. To achieve this, we define the **interleaved multimodal input** format to feed videos and subtitles together into LMMs as temporally-aligned multimodal sequences. Specifically, a chunk of subtitle will be inserted in-between the two frames before and after the mid-timestamp of the subtitle.

## 3.3 Annotating Questions and Answers

We conduct the annotation process in a well-controlled lab environment with experienced annotators. Before annotation, we conduct a special training to all annotators for them to understand the requirements of each specific question category. During the annotation process, the subtitles are appended at the bottom of the video with aligned timestamps, and displayed to annotators. The annotator is required to watch the full video before starting the annotation, and is allowed to drag back to to any specific timestamps after full watching. We collect one question per video for videos longer than 60 seconds, two questions for videos in *(180s, 600s]*, three questions for *(900s, 3600s]*. The annotator also needs to provide 3-4 distracting answer options that are relevant to the question and the video.

We further introduce two additional annotation requirements to ensure high-quality *referred reasoning* questions: 1) We explicitly require annotators to include and highlight the referred query in all questions[2]; 2) To ensure that the referred context uniformly span over the video, we ask annotators to explicitly label the frame index for all referred moments in each question. This additional requirement further facilitates our in-depth study of long-context understanding abilities for LMMs with respect to the relative token-wise distance between the question and the referred context.

To control the annotation quality, each video is passed through three annotators: 1) The primary annotator, whose duty is to provide annotations and filter out videos that are not available for annotation (*e.g.* still frames, incomplete subtitles); 2) The examiner, who examines whether the annotated question is in the correct question category, and whether the annotation requirements are all met; 3) The reviser, to revise the annotations labeled as incorrect by examiners. The examiner and reviser have identified 20% of annotations to be problematic and revised them, which significantly improved the quality of the LONGVIDEOBENCH.

As we require all questions to include the question body itself as well as a referring query, the average question length is as long as **43.53** words, ensuring that the referred context is clearly depicted in the question without introducing ambiguity. The average length of an answer is 8.28 words.

## 4 Evaluation of LONGVIDEOBENCH

### 4.1 Models and Evaluation Strategies

**Participating LMMs.** We include in total 22 LMMs for evaluation. The main participants are long-context LMMs, including four proprietary models: GPT-4o (gpt-4o-0513), Gemini-1.5-Pro (gemini-1.5-pro-0514), GPT-4-Turbo (gpt-4-turbo-0409), and Gemini-1.5-Flash (gemini-1.5-flash-0514), and four state-of-the-art open-sourced long-context LMMs: Phi-3-Vision-Instruct (*128K*), Idefics2 (*32K*), Mantis-Idefics2 (*32K*), and Mantis-BakLLaVA (*32K*). All these models above support interleaved video-language inputs. We also evaluate 9 representative video-specific LMMs, and 6 image LMMs that support $\geq 8$ images.

**Validation and Test Subsets.** We split the LONGVIDEOBENCH into two subsets, the *validation set* (752 videos, 1337 MCQs), and the *test set* (3011 videos, 5341 MCQs). We use the *validation set* to analyze the performance of LMMs under different settings. Afterwards, we pick the optimal setting for each LMM to report their performance on test set leaderboard.

### 4.2 Main Results

In Tab. 5 and Tab. 6, we analyze the performance of six long-context LMMs under different settings on the val set of LONGVIDEOBENCH. Our evaluation brings several important findings, as follows:

**1) *LMMs have to understand long inputs for better results.*** As shown in Tab. 5 (a), (b), (c) and (d), all four proprietary models, especially more advanced GPT-4o and Gemini-Pro, have shown significant improvements while increasing their input length, in particular for long videos. For videos longer than 180 seconds, GPT-4o and Gemini-1.5-Pro can improve more than **10%** by increasing input length from 16 to 256 frames. In contrast, on EgoSchema, Gemini-1.5-Pro only improves 2.5% from 16 to 150 frames. This validates the effectiveness of LONGVIDEOBENCH as a longstanding challenging benchmark for models to evaluate their long-context multimodal understanding abilities.

**2) *Open-source models lag significantly behind.*** Different from proprietary models, open-source LMMs are unable to improve their results by inputting more than 16 frames. Idefics2 and Mantis-Idefics2, as shown in Tab. 5 (e) and (g), even face a severe degradation on accuracy with 64 input frames, before they have reached their context length limits.

**3) *Longer videos are more challenging.*** As in Tab. 5, all six models show the lowest accuracy on the longest *(900,3600]* group, followed by the *(180,600]* group, and then the shorter-video groups. These results pose LONGVIDEOBENCH as a meaningful and challenging benchmark for LMMs to test their video understanding abilities.

---

[2]Except SEQUENCE OF SCENES questions, where it is *implicitly* mentioned in all candidate choices.

Table 5: Validation set results categorized by duration groups, *w.r.t.* `max_frames` (capped at 1 *fps*). While `max_frames` is already more than the max duration of a group, the results will not change when we set a larger `max_frames`. Respective settings are labeled as "s.a." (same as above).

| Model | max_frames | Duration Group (unit: second) (8,15] | (15,60] | (180,600] | (900,3600] | all | Model | max_frames | Duration Group (unit: second) (8,15] | (15,60] | (180,600] | (900,3600] | all |
|---|---|---|---|---|---|---|---|---|---|---|---|---|---|
| (a) GPT-4O | 1 | 52.9 | 50.6 | 40.8 | 36.0 | 41.7 | (b) GEMINI-1.5-PRO | 1 | 46.6 | 45.2 | 35.7 | 35.8 | 38.6 |
| | 4 | 63.5 | 64.3 | 47.2 | 40.3 | 48.7 | | 4 | 59.6 | 62.9 | 37.7 | 39.0 | 44.9 |
| | 8 | 69.7 | 67.3 | 49.4 | 47.1 | 53.3 | | 8 | 62.4 | 68.0 | 44.9 | 46.0 | 51.0 |
| | 16 | **71.4** | 73.7 | 53.8 | 52.2 | 58.0 | | 16 | **67.4** | 69.6 | 50.3 | 44.0 | 52.7 |
| | 32 | s.a. | 73.5 | 57.3 | 50.5 | 58.5 | | 32 | s.a. | 74.3 | 51.2 | 48.0 | 55.2 |
| | 64 | s.a. | **76.7** | 61.4 | 55.8 | 62.0 | | 64 | s.a. | **75.1** | 59.3 | 50.9 | 58.6 |
| | 128 | s.a. | s.a. | 64.2 | 56.5 | 63.5 | | 128 | s.a. | s.a. | 64.9 | 54.0 | 61.9 |
| | 256 | s.a. | s.a. | **69.1** | **60.9** | **66.7** | | 256 | s.a. | s.a. | **65.3** | **58.6** | **64.0** |
| (c) GPT-4-TURBO | 1 | 49.2 | 48.3 | 43.7 | 39.2 | 43.2 | (d) GEMINI-1.5-FLASH | 1 | 48.6 | 42.9 | 35.1 | 35.4 | 38.1 |
| | 4 | 57.1 | 57.0 | 46.6 | 43.8 | 48.1 | | 4 | 53.3 | 64.5 | 40.0 | 40.4 | 45.2 |
| | 8 | 59.8 | 62.8 | 50.7 | 41.5 | 49.7 | | 8 | 62.5 | 65.3 | 45.8 | 41.8 | 48.9 |
| | 16 | **65.2** | 67.9 | 51.7 | 44.5 | 52.7 | | 16 | **68.3** | 66.9 | 49.0 | 43.9 | 50.8 |
| | 32 | s.a. | 66.9 | 53.1 | 47.5 | 54.0 | | 32 | s.a. | 74.1 | 50.0 | 44.5 | 53.5 |
| | 64 | s.a. | **68.2** | 59.0 | 47.0 | 56.0 | | 64 | s.a. | **76.2** | 54.4 | 48.6 | 56.8 |
| | 128 | s.a. | s.a. | 60.3 | 49.3 | 57.5 | | 128 | s.a. | s.a. | 56.9 | 51.7 | 58.9 |
| | 256 | s.a. | s.a. | **62.4** | **50.5** | **59.0** | | 256 | s.a. | s.a. | **62.6** | **54.0** | **61.6** |

| Model | max_frames | Duration Group (unit: second) (8,15] | (15,60] | (180,600] | (900,3600] | all | Model | max_frames | Duration Group (unit: second) (8,15] | (15,60] | (180,600] | (900,3600] | all |
|---|---|---|---|---|---|---|---|---|---|---|---|---|---|
| (e) IDEFICS2 | 1 | 48.6 | 48.8 | 39.3 | 38.5 | 41.5 | (f) PHI-3-VISION | 1 | 49.2 | 46.5 | 39.3 | 37.4 | 40.8 |
| | 4 | 62.4 | 58.1 | 41.3 | 41.3 | 46.4 | | 4 | 56.6 | 57.5 | 44.4 | 43.6 | 47.5 |
| | 8 | 59.3 | 63.4 | 46.8 | 41.7 | 48.5 | | 8 | 60.8 | 62.2 | 42.5 | 43.6 | 48.1 |
| | 16 | **59.8** | **65.7** | **47.8** | **42.7** | **49.7** | | 16 | **59.3** | 61.6 | **46.8** | **44.7** | **49.6** |
| | 32 | s.a. | 64.5 | 44.0 | 41.5 | 47.8 | | 32 | s.a. | **66.3** | 46.6 | 42.3 | 49.1 |
| | 64 | s.a. | 52.3 | 22.1 | 21.2 | 30.9 | | 64 | – Context Length Exceeded – | | | | |

| Model | max_frames | Duration Group (unit: second) (8,15] | (15,60] | (180,600] | (900,3600] | all | Model | max_frames | Duration Group (unit: second) (8,15] | (15,60] | (180,600] | (900,3600] | all |
|---|---|---|---|---|---|---|---|---|---|---|---|---|---|
| (g) MANTIS-IDEFICS2 | 1 | 48.1 | 44.2 | 35.4 | 36.2 | 38.7 | (h) MANTIS-BAK LLAVA | 1 | 48.1 | 44.2 | 35.4 | 36.2 | 38.7 |
| | 4 | 53.4 | 51.2 | 42.5 | 38.7 | 43.5 | | 4 | **57.7** | 50.0 | 38.8 | 36.5 | 42.0 |
| | 8 | **57.7** | **57.0** | 45.4 | 39.5 | 46.1 | | 8 | 54.0 | 55.8 | 39.8 | 37.8 | 43.0 |
| | 16 | 56.6 | 55.8 | **45.6** | **42.2** | **47.0** | | 16 | 53.4 | **57.6** | **40.3** | **38.7** | **43.7** |
| | 32 | s.a. | 55.8 | 42.7 | 40.4 | 45.4 | | 32 | s.a. | 54.7 | 39.8 | 37.8 | 42.8 |
| | 64 | s.a. | 48.0 | 24.7 | 24.9 | 30.2 | | 64 | – Context Length Exceeded – | | | | |

Table 6: Validation set results *w.r.t.* input modalities.

| Video Frames? | Text Subtitles? | GPT-4O | GEMINI-1.5-PRO | GPT-4-TURBO | GEMINI-1.5-FLASH | IDEFICS2 | PHI-3-VISION | MANTIS-IDEFICS2 | MANTIS-BAKLLAVA |
|---|---|---|---|---|---|---|---|---|---|
| ✗ | ✓ | 44.6 | 43.0 | 45.2 | 39.2 | 25.6 | 40.7 | 31.7 | 31.1 |
| ✓ | ✗ | 60.6 | 62.9 | 56.0 | 60.2 | 49.4 | 49.5 | 45.8 | 43.5 |
| ✓ | ✓ | **66.7** | **63.9** | **59.0** | **61.6** | **49.7** | **49.6** | **47.0** | **43.7** |

**4) *Interleaved inputs are hard.*** As shown in Tab. 6, all models can improve their results by inserting subtitles to videos as inputs. However, compared to GPT-4o, open-source LMMs are still unable to effectively integrate subtitle information to facilitate video understanding and improve their accuracy on LONGVIDEOBENCH, demonstrating a gap in long-context multimodal understanding.

**5) *Visual modality is fundamental.*** Results from Tab. 6 also demonstrate that video frames, *i.e.* visual modality, is a crucial component in the interleaved inputs, as removing them and only using the subtitles lead to much worse results for all models.

## 4.3 Leaderboard

Table 7 shows the test set results of the 6 long-context LMMs, as well as 9 representative open-source video LMMs and 6 open-source image LMMs with multi-image support. By including more LMMs for evaluation, this leaderboard raises more observations, as follows:

**6) *Open-source video LMMs do not show clear advantages over image LMMs.*** Under the same model architecture, LLaVA-Next-Video-M7B (*video LMM*) is less competitive than LLaVA-Next-Mistral (*image LMM*), despite being trained on additional videos. This may be due to existing video training datasets mainly consisting of short videos and summary-level tasks, leading to a decline on long-context and detailed video understanding capabilities.

Table 7: Test Set Leaderboard of the LONGVIDEOBENCH on 23 LMMs up to submission, with more recent results in Tab. A. We show the validation set results ('Val Total') as a reference.

| Model | Val Total | Duration Group (s) | | | | Question Category | | | | | | | | | | | | | | | | | Test Total |
|---|---|---|---|---|---|---|---|---|---|---|---|---|---|---|---|---|---|---|---|---|---|---|---|
| | | (8, 15] | (15, 60] | (180, 600] | (900, 3600] | (L1) Perception | | | | | | | | (L2) Relation | | | | | | | | | |
| | | | | | | S2E | S2O | S2A | E2O | O2E | T2E | T2O | T2A | E3E | O3O | SSS | SOS | SAA | T3E | T3O | TOS | TAA | |
| *Proprietary Long-context LMMs: (max_frames set according to Tab. 5)* | | | | | | | | | | | | | | | | | | | | | | | |
| GPT-4o (0513) | 66.7 | 71.6 | 76.8 | 66.7 | 61.6 | 76.8 | 69.8 | 70.9 | 67.3 | 72.8 | 67.2 | 65.3 | 77.2 | 62.6 | 61.3 | 44.3 | 75.6 | 62.6 | 64.0 | 66.4 | 62.1 | 66.4 | 66.7 |
| Gemini-1.5-Pro (0514) | 64.0 | 70.2 | 75.3 | 65.0 | 59.1 | 74.6 | 58.3 | 76.2 | 68.7 | 73.3 | 66.2 | 63.6 | 76.7 | 61.9 | 58.6 | 55.2 | 69.0 | 59.0 | 58.9 | 60.5 | 53.3 | 62.5 | 64.4 |
| Gemini-1.5-Flash (0514) | 61.6 | 66.1 | 73.1 | 63.1 | 57.3 | 68.5 | 64.7 | 68.0 | 64.5 | 72.5 | 63.6 | 68.0 | 76.7 | 56.5 | 61.0 | 43.1 | 67.3 | 56.2 | 57.5 | 55.0 | 55.3 | 60.7 | 62.4 |
| GPT-4-Turbo (0409) | 59.1 | 66.4 | 71.1 | 61.7 | 54.5 | 74.9 | 60.1 | 64.2 | 63.9 | 69.4 | 62.5 | 61.3 | 69.9 | 57.5 | 55.9 | 44.8 | 66.0 | 53.2 | 56.5 | 53.6 | 56.2 | 60.2 | 60.7 |
| *Open-source Long-Context LMMs: (max_frames set according to Tab. 5)* | | | | | | | | | | | | | | | | | | | | | | | |
| Idefics2 | 49.7 | 57.4 | 60.4 | 47.3 | 44.7 | 60.9 | 51.4 | 49.4 | 53.7 | 58.9 | 54.4 | 51.8 | 54.8 | 46.8 | 40.5 | 28.9 | 61.0 | 49.8 | 47.0 | 42.0 | 40.7 | 46.2 | 49.4 |
| Phi-3-Vision-Instruct | 49.6 | 58.3 | 59.6 | 48.4 | 45.1 | 60.3 | 52.9 | 53.4 | 51.8 | 54.1 | 52.3 | 55.3 | 53.3 | 49.4 | 47.6 | 33.6 | 59.3 | 46.2 | 44.2 | 43.2 | 38.8 | 51.5 | 49.9 |
| Mantis-Idefics2 | 47.0 | 56.1 | 61.4 | 44.6 | 42.5 | 60.3 | 51.1 | 51.2 | 53.4 | 52.9 | 51.4 | 49.5 | 57.3 | 46.2 | 45.1 | 30.2 | 53.7 | 46.5 | 44.2 | 40.1 | 30.6 | 40.2 | 47.6 |
| Mantis-BakLLaVA | 43.7 | 51.3 | 52.7 | 41.1 | 40.1 | 53.0 | 38.7 | 44.1 | 46.0 | 51.0 | 50.8 | 43.7 | 50.8 | 45.5 | 40.2 | 23.3 | 48.0 | 44.9 | 40.9 | 38.5 | 34.9 | 47.7 | 43.7 |
| *Open-source Image LMMs with Multi-Image Support: (all sample 8 frames)* | | | | | | | | | | | | | | | | | | | | | | | |
| LLaVA-Next-Mistral-7B | 49.1 | 53.4 | 57.2 | 46.9 | 42.1 | 59.0 | 46.5 | 49.4 | 49.7 | 52.2 | 52.9 | 51.1 | 51.4 | 47.4 | 45.4 | 28.2 | 56.0 | 50.8 | 38.7 | 41.6 | 31.9 | 48.1 | 47.1 |
| InstructBLIP-T5-XXL | 43.3 | 48.1 | 50.1 | 44.5 | 40.0 | 54.9 | 39.3 | 41.3 | 45.4 | 49.7 | 52.9 | 42.4 | 48.6 | 44.2 | 40.2 | 25.2 | 51.0 | 42.9 | 42.7 | 41.6 | 33.9 | 47.7 | 43.8 |
| BLIP-2-T5-XXL | 42.7 | 46.7 | 47.4 | 44.2 | 40.9 | 54.6 | 38.1 | 38.8 | 46.3 | 49.0 | 52.6 | 40.2 | 44.3 | 45.2 | 41.2 | 25.6 | 51.3 | 41.6 | 45.1 | 45.1 | 33.6 | 47.4 | 43.5 |
| LLaVA-1.5-13B | 43.4 | 49.0 | 51.1 | 41.8 | 39.6 | 54.9 | 42.6 | 40.4 | 44.8 | 49.9 | 51.1 | 43.1 | 43.0 | 45.2 | 40.9 | 29.9 | 53.3 | 44.2 | 38.7 | 35.6 | 30.0 | 46.2 | 43.1 |
| LLaVA-1.5-7B | 40.3 | 45.0 | 47.4 | 40.1 | 37.0 | 53.3 | 35.0 | 38.8 | 39.6 | 44.9 | 44.1 | 39.9 | 43.3 | 40.7 | 43.9 | 26.2 | 47.3 | 42.9 | 37.2 | 34.7 | 30.3 | 45.1 | 40.4 |
| mPLUG-Owl2 | 39.1 | 49.4 | 47.3 | 38.7 | 34.3 | 49.5 | 37.5 | 37.3 | 39.6 | 45.5 | 45.9 | 41.5 | 39.6 | 44.6 | 36.9 | 24.9 | 45.7 | 38.9 | 30.9 | 36.6 | 33.9 | 38.3 | 39.4 |
| *Open-source Video LMMs: (frame sampling set as their default settings)* | | | | | | | | | | | | | | | | | | | | | | | |
| PLLaVA-34B | 53.2 | 60.1 | 66.8 | 50.8 | 49.1 | 65.9 | 53.8 | 53.1 | 54.9 | 57.6 | 58.9 | 52.4 | 56.3 | 54.8 | 50.6 | 44.2 | 60.3 | 56.1 | 46.6 | 47.9 | 41.4 | 54.9 | 53.5 |
| LLaVA-Next-Video-34B | 50.5 | 57.6 | 61.6 | 48.7 | 45.9 | 62.1 | 50.2 | 51.2 | 50.9 | 58.5 | 59.0 | 48.2 | 48.9 | 54.8 | 49.7 | 39.2 | 58.7 | 50.8 | 46.6 | 43.8 | 36.8 | 47.2 | 50.5 |
| PLLaVA-13B | 45.6 | 52.9 | 54.3 | 42.9 | 41.2 | 57.1 | 43.5 | 41.9 | 43.5 | 53.5 | 54.4 | 46.9 | 43.7 | 47.1 | 43.6 | 27.2 | 58.0 | 54.4 | 39.6 | 40.1 | 30.9 | 47.0 | 45.1 |
| LLaVA-Next-Video-M7B | 43.5 | 50.9 | 53.1 | 42.6 | 38.9 | 54.6 | 41.7 | 47.2 | 46.3 | 52.9 | 46.8 | 46.6 | 45.8 | 44.9 | 42.1 | 24.6 | 51.3 | 40.6 | 39.0 | 40.1 | 34.5 | 39.5 | 43.5 |
| ShareGPT4Video | 39.7 | 46.9 | 50.1 | 40.0 | 38.7 | 50.2 | 37.5 | 44.4 | 44.2 | 42.7 | 43.8 | 41.2 | 45.8 | 41.7 | 42.7 | 29.9 | 50.3 | 47.2 | 38.7 | 39.7 | 29.3 | 39.8 | 41.8 |
| PLLaVA-7B | 40.2 | 45.3 | 47.3 | 38.5 | 35.2 | 52.4 | 35.3 | 40.4 | 39.3 | 46.8 | 46.5 | 39.9 | 39.3 | 41.0 | 36.3 | 26.2 | 47.7 | 41.6 | 34.1 | 30.5 | 27.7 | 38.3 | 39.2 |
| VideoChat2 (Mistral-7B) | 39.3 | 49.3 | 49.3 | 39.0 | 37.5 | 53.6 | 40.8 | 38.5 | 44.5 | 53.5 | 46.8 | 43.1 | 47.7 | 43.6 | 46.6 | 10.6 | 42.0 | 40.6 | 38.4 | 36.3 | 27.4 | 43.6 | 41.2 |
| VideoLLaVA | 39.1 | 43.1 | 44.6 | 36.4 | 34.4 | 49.5 | 29.6 | 30.6 | 40.9 | 44.9 | 43.5 | 33.8 | 40.6 | 46.5 | 38.7 | 24.3 | 40.0 | 42.9 | 35.1 | 30.5 | 23.8 | 39.5 | 37.6 |
| VideoChat2 (Vicuna 7B) | 36.0 | 38.1 | 40.5 | 33.5 | 33.6 | 44.8 | 29.0 | 27.3 | 36.9 | 41.7 | 41.7 | 34.1 | 33.1 | 37.2 | 39.6 | 22.6 | 43.0 | 30.7 | 34.1 | 33.8 | 28.3 | 37.2 | 35.1 |

**7) *Stronger LLM backbones are helpful.*** Compared with PLLaVA-7B, its larger variants trained with the same datasets, PLLaVA-13B and PLLaVA-34B, shows notable 5.9% and 14.3% improvements, and PLLaVA-34B ranks top among all open-source models. This observation suggests that scaling up the language model is effective for more comprehensive video understanding.

**8) *(L2) Relation is more challenging than (L1) Perception.*** Compared to (L1), questions in (L2) additionally require LMMs to understand the relation among multiple scenes in the video. Thus, the disparity between performance on (L1) and (L2) indicates LMMs' insufficient understanding of the temporal dynamics of videos. The most difficult category is an (L2) category, SSS (SEQUENCE OF SCENES), where the distracting options are permutations of the correct sequence order (of the scenes). All LMMs perform worst on this category of questions, further highlighting their limitation of complex temporal understanding.

**9) *Results on validation and test subsets are consistent.*** This consistency demonstrates the validation set as a sufficient representation of the entire benchmark dataset, confirming the reliability of LONGVIDEOBENCH and the findings in Sec. 4.2.

### 4.4 Performance *w.r.t.* Referring Query Depth

In Fig. 4, we further analyze the performance trends of LMMs when the queried moment is located at different positions within a video. In summary, the performance of LMMs is not uniform: all models perform worse when the referred moment is closer to the beginning of the video (*i.e.* has longer distance to the question), and this trend becomes more evident as the video duration becomes longer. Additionally, we found that questions posed closer to the middle of the video, rather than the beginning or end, present a greater challenge for LMMs. These findings are consistent with respective conclusions from needle in a haystack (NIAH) [gkamradt, 2024] for long-term text understanding.

## 5 Related Works

**Video LMMs and Long-context LMMs.** Early video LMMs focus on short videos (less than one minute). These works usually build upon pre-trained video backbones [Wang et al., 2023, 2024b], temporal pooling modules [Zhang et al., 2023b,c, Xu et al., 2024] and are trained on video-specific supervised tuning datasets [Li et al., 2023a, Zhang et al., 2023c]. Several image LMMs [Li et al.,

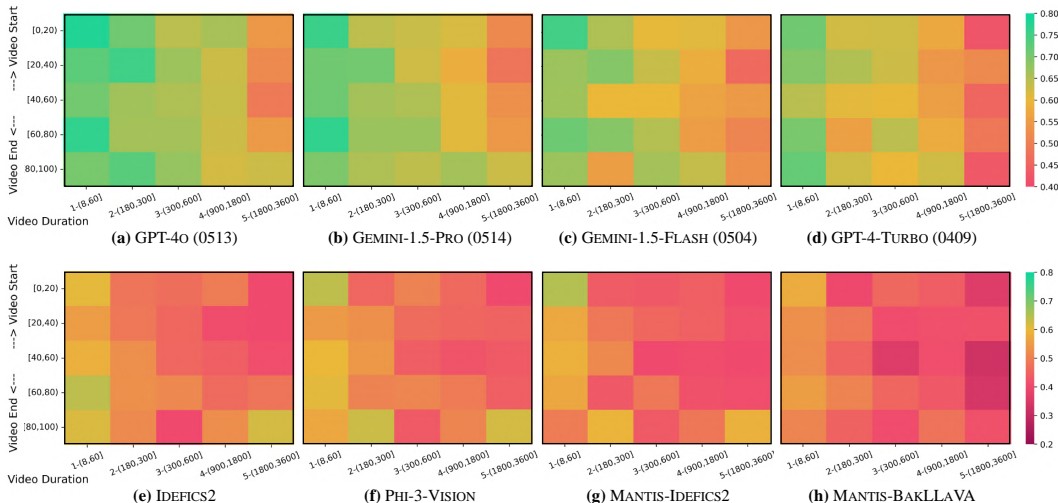

Figure 4: Accuracy of proprietary and open-source LMMs *w.r.t.* referring query depth and video duration. All models perform worse when the referred moment is closer to video start or middle video. Please refer to Appendix Sec. C for respective visualizations on rest 15 models.

2023b, Liu et al., 2023a, Ye et al., 2023] have shown competitive performance on many traditional short-video understanding tasks [Xu et al., 2017, Yu et al., 2019].

For longer videos, recent research explores methods like compressing video frames to fewer tokens to manage hour-long content within LMMs [Li et al., 2023c], and incorporating memory banks into standard LMM architectures [Song et al., 2023, He et al., 2024, Tan et al., 2024]. Leading models, both open-source (*e.g.,* LWM [Liu et al., 2024a], Phi-3-128K [Abdin et al., 2024]) and proprietary (*e.g.,* GPT-4o [OpenAI, 2024a], Gemini-1.5-Pro [Team, 2024]), now support context lengths over 128K tokens, allowing detailed video analysis. However, robust benchmarks for long-duration video understanding are lacking, with GPT-4o assessed only on 3-minute videos [Yu et al., 2019, Mangalam et al., 2023] and Gemini-1.5-Pro on an in-house benchmark. To advance LMM capabilities in understanding longer videos, we introduce LONGVIDEOBENCH, a comprehensive benchmark for evaluating LMMs across various video durations and distributions.

**Benchmarks for Video LMMs.** Traditionally, video LMMs are evaluated on classical video QA datasets like MSVD-QA, MSRVTT-QA [Xu et al., 2017], and ActivityNet-QA [Yu et al., 2019], which primarily evaluate video LMMs through global-summary questions. However, it has been demonstrated that these benchmarks are addressable by a few key frames. For a focused assessment of temporal comprehension, NeXT-QA [Xiao et al., 2021] and MVBench [Wang et al., 2023] serve to measure temporal dynamics over short clips, with average durations of *44s* and *16s*, respectively. Long-duration video understanding is targeted by benchmarks like EgoSchema [Mangalam et al., 2023], which involves multi-choice questions on *3-minute-long* egocentric videos, and MovieChat-1K [Song et al., 2023], focused on *10-minute-long* movie clips. These long-video benchmarks often limit their scope to videos on specific themes and still include a large proportion of summary questions solvable with limited frames. To address these gaps and enhance evaluation of detailed multimodal reasoning over longer videos, we introduce the LONGVIDEOBENCH, a comprehensive benchmark focusing on referring reasoning questions that by-design requires dense input frames to solve, encompassing diverse video topics and varying lengths up to hour long.

## 6 Conclusion

This work introduces LONGVIDEOBENCH, a comprehensive benchmark that evaluates Large Multi-modal Models (LMMs) in understanding hour-long subtitled videos in diverse themes. The benchmark introduces referring reasoning questions, a novel video question-answering paradigm that addresses the longstanding issue of single frame bias in existing video understanding benchmarks. Evaluation results demonstrate that LONGVIDEOBENCH presents significant challenges for both proprietary and open-source LMMs in their long-context multimodal capabilities. In addition, the benchmark results provide valuable insights on the deficiencies of existing models, making it a valuable asset to understand the current multimodal model landscape and to guide the future explorations.

## Acknowledgement

We thank our colleagues for their helpful discussions. We appreciate conference committee chairs and members for their voluntary services and valuable feedback. We are grateful for authors who provide open or API access to their models for the evaluation.

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

## A  Updated Leaderboard with Newer Models

Since the initial release of the benchmark, multiple models (Aria, LLaVA-OneVision, *etc*) have submitted their results to LONGVIDEOBENCH. In addition, we evaluate several up-to-date popular models with their default / recommended settings. The updated leaderboard until October 30th, 2024 is presented in Table 8.

Table 8: Updated leaderboard until Oct 30th, 2024, with performance of 35 models available. Live-updating leaderboard is on `https://longvideobench.github.io`.

| Model | Open? | max_frames | Test Total | (8,15] | (15, 60] | (180,600] | (900,3600] | Val Total |
|---|---|---|---|---|---|---|---|---|
| GPT-4O (0513) | ✗ | 256 | 66.7 | 71.6 | 76.8 | 66.7 | 61.6 | 66.7 |
| ARIA (MoE-3.5A-25B) | ✓ | 256 | 65.0 | 69.4 | 76.6 | 64.6 | 60.1 | 64.2 |
| LLAVA-VIDEO-72B-QWEN2 | ✓ | 128 | 64.9 | 72.4 | 77.4 | 63.9 | 59.3 | 63.9 |
| GEMINI-1.5-PRO (0514) | ✗ | 256 | 64.4 | 70.2 | 75.3 | 65.0 | 59.1 | 64.0 |
| LLAVA-ONEVISION-72B | ✓ | 32 | 63.2 | 74.3 | 77.4 | 61.6 | 56.5 | 61.3 |
| LLAVA-VIDEO-7B-QWEN2 | ✓ | 128 | 62.7 | 69.7 | 76.5 | 62.1 | 56.6 | 61.1 |
| GEMINI-1.5-FLASH (0514) | ✗ | 256 | 62.4 | 66.1 | 73.1 | 63.1 | 57.3 | 61.6 |
| GPT-4-TURBO (0409) | ✗ | 256 | 60.7 | 66.4 | 71.1 | 61.7 | 54.5 | 59.1 |
| INTERNVL2-40B | ✓ | 16 | 60.6 | 71.4 | 76.6 | 57.5 | 54.4 | 59.3 |
| GPT-4O-MINI (0718) | ✗ | 250 | 58.8 | 66.6 | 73.4 | 56.9 | 53.4 | 56.5 |
| MINICPM-V-2.6 | ✓ | 64 | 57.7 | 62.7 | 69.1 | 54.9 | 49.8 | 54.9 |
| QWEN2-VL-7B | ✓ | 256 | 56.8 | 60.1 | 67.6 | 56.7 | 52.5 | 55.6 |
| KANGAROO | ✓ | 64 | 54.8 | 61.5 | 65.7 | 52.7 | 49.1 | 54.2 |
| PLLAVA-34B | ✓ | 32 | 53.5 | 60.1 | 66.8 | 50.8 | 49.1 | 53.2 |
| INTERNVL-CHAT-V1.5-26B | ✓ | 16 | 51.7 | 61.3 | 62.7 | 49.5 | 46.6 | 51.2 |
| LLAVA-NEXT-VIDEO-34B | ✓ | 32 | 50.5 | 57.6 | 61.6 | 48.7 | 45.9 | 50.5 |
| PHI-3-VISION-INSTRUCT | ✓ | 16 | 49.9 | 58.3 | 59.6 | 48.4 | 45.1 | 49.6 |
| IDEFICS2 | ✓ | 16 | 49.4 | 57.4 | 60.4 | 47.3 | 44.7 | 49.7 |
| MANTIS-IDEFICS2 | ✓ | 16 | 47.6 | 56.1 | 61.4 | 44.6 | 42.5 | 47.0 |
| PIXTRAL-12B | ✓ | 16 | 47.4 | 51.3 | 55.2 | 45.7 | 44.4 | 44.9 |
| LLAVA-NEXT-MISTRAL-7B | ✓ | 8 | 47.1 | 53.4 | 57.2 | 46.9 | 42.1 | 49.1 |
| LLAMA3.2-11B-VISION | ✓ | 16 | 45.7 | 52.3 | 53.7 | 43.9 | 42.4 | 45.4 |
| PLLAVA-13B | ✓ | 32 | 45.1 | 52.9 | 54.3 | 42.9 | 41.2 | 45.6 |
| INSTRUCTBLIP-T5-XXL | ✓ | 8 | 43.8 | 48.1 | 50.1 | 44.5 | 40.0 | 43.3 |
| MANTIS-BAKLLAVA | ✓ | 16 | 43.7 | 51.3 | 52.7 | 41.1 | 40.1 | 43.7 |
| BLIP-2-T5-XXL | ✓ | 8 | 43.5 | 46.7 | 47.4 | 44.2 | 40.9 | 42.7 |
| LLAVA-NEXT-VIDEO-M7B | ✓ | 32 | 43.5 | 50.9 | 53.1 | 42.6 | 38.9 | 43.5 |
| LLAVA-1.5-13B | ✓ | 8 | 43.1 | 49.0 | 51.1 | 41.8 | 39.6 | 43.4 |
| SHAREGPT4VIDEO | ✓ | 16 | 41.8 | 46.9 | 50.1 | 40.0 | 38.7 | 39.7 |
| VIDEOCHAT2 (MISTRAL-7B) | ✓ | 16 | 41.2 | 49.3 | 49.3 | 39.0 | 37.5 | 39.3 |
| LLAVA-1.5-7B | ✓ | 8 | 40.4 | 45.0 | 47.4 | 40.1 | 37.0 | 40.3 |
| MPLUG-OWL2 | ✓ | 8 | 39.4 | 49.4 | 47.3 | 38.7 | 34.3 | 39.1 |
| PLLAVA-7B | ✓ | 32 | 39.2 | 45.3 | 47.3 | 38.5 | 35.2 | 40.2 |
| VIDEOLLAVA | ✓ | 8 | 37.6 | 43.1 | 44.6 | 36.4 | 34.4 | 39.1 |
| VIDEOCHAT2 (VICUNA 7B) | ✓ | 16 | 35.1 | 38.1 | 40.5 | 33.5 | 33.6 | 36.0 |

# B  Additional Experimental Settings

**Resources.** All experiments on open-source LMMs are conducted on cloud servers with 2*NVIDIA A100 80G GPUs and 64-core Intel(R) Xeon(R) Platinum 8336C CPUs. For GPT-4o and Gemini-1.5-Pro, we obtain their results where their API services. To avoid unknown impacts on results, we do not conduct batched inference and set *batch size=1* for all 18 participating open-source LMMs.

## B.1  A Brief Introduction on Participating Models

We introduce the 6 participating long-context LMMs as follows:

*GPT-4o.* Released on May 13, 2024 with an API service, GPT-4o is the latest model among multimodal LMMs provided by OpenAI. It has 128K context length.

*Gemini-1.5-Pro.* Released on May 14, 2024 with an API service, the Gemini-1.5-Pro version that we have evaluated is the upgraded version of its Febrary release. It has 10M context length.

*Gemini-1.5-Flash.* Released on May 14, 2024 with an API service, the Gemini-1.5-Flash version that we have evaluated is the upgraded version of its Febrary release. It has 10M context length.

*GPT-4-Turbo.* Released on April 9, 2024 with an API service, GPT-4o is the earlier multimodal LMM provided by OpenAI. It has 128K context length.

*Idefics2.* Released on Apr 15, 2024, Idefics2 [Laurençon et al., 2024] is the latest open-source LMM released by Hugging Face, with total 8B parameters, as an improved version of Idefics [Huggingface, 2023]. Initializing from the Mistral-Instruct-v0.2 [Jiang et al., 2023], it has 32K context length.

*Phi-3-Vision-Instruct.* Released on May 22, 2024, Phi-3-Vision-Instruct [Abdin et al., 2024] an open-source long-context LMM released by Microsoft. It has 4B parameters and 128K context length. As each image consumes over 2000 tokens in it, Phi-3-Vision-Instruct does not support settings with 64 or more input frames in Tab. 5.

*Mantis-Idefics2 and Mantis-BakLLaVA.* Mantis [Jiang et al., 2024] is an academic attempt on improving multi-image and video understanding abilites of LMMs. It proposes the Mantis-Instruct dataset, a re-purpose of open-access multi-image and video datasets, and fine-tunes over existing LMMs: Idefics2 and BakLLaVA [SkunkworksAI, 2024], into Mantis-Idefics2 and Mantis-BakLLaVA. While both Mantis have 32K context length, the Mantis-BakLLaVA requires 576 tokens per frame, and thus, same as Phi-3-Vision-Instruct, it does not support settings with 64 or more input frames.

## B.2  Prompts and Settings

**Group 1: Long-context LMMs.** For the 6 long-context LMMs (GPT-4o, Gemini-1.5-Pro, Idefics2, Phi-3, Mantis-Idefic2 and Mantis-BakLLaVA), as they all support free-composition image-text interleaved inputs, our inputs to each of them is a list of images (or image placeholders) and texts, ending with the question and all options (images labeled in blue):

```
<image>
<image>
<subtitle>
<image>
<subtitle>
...
Question:  <Question>
<Options>
Answer with the option's letter from the given choices directly.
```

Specifically, the `<image>` are `PIL.Image` objects, and other text elements are strings.

*GPT-4o & GPT-4-Turbo.* The image and text elements are converted as follows before requesting the API service.

```
# image elements
image_elem = {
            "type": "image_url",
            "image_url": {
                "url": f"data:image/jpeg;base64,{base64_image}",
            },
        }

# text elements
text_elem = {
            "type": "text",
            "text": text,
}
```

*Gemini-1.5-Pro & Gemini-1.5-Flash.* Respective video frames are pre-extracted and uploaded to temporary links on Google Cloud Storage (GCS). Then, images are requested with these GCS links.

```
from vertexai.generative_models import Part

# image elements
image_elem = Part.from_uri(
            gcs_path,
            mime_type="image/jpeg",
```

```
    )

    # text elements
    text_elem = text
```

*Idefics2.* We define the pre-processing function for Idefics2 from our raw interleaved list to its inputs:

```
def idefics2_preprocess(interleaved_list):
    placeholder_list, image_list = [], []
    for itm in interleaved_list:
        if isinstance(itm, str):
            placeholder_list.append({"type": "text", "text": itm})
        else:
            placeholder_list.append({"type": "image"})
            image_list.append(itm)
    return placeholder_list, image_list
```

*Phi-3.* The pre-processing for Phi-3 is as follows:

```
class Counter:
    def __init__(self):
        self.counter = 0

    def __call__(self):
        self.counter += 1
        return self.counter

user_prompt = "<|user|>"
assistant_prompt = "<|assistant|>"
prompt_suffix = "<|end|>"

def phi3_preprocess(interleaved_list):
    counter = Counter()
    placeholder_list, image_list = [user_prompt], []
    for itm in interleaved_list:
        if isinstance(itm, str):
            placeholder_list.append(itm)
        else:
            placeholder_list.append("<|image_{}|>".format(counter()))
            image_list.append(resize_image(itm))
    placeholder_list.extend([prompt_suffix, assistant_prompt, ""])
    return "\n".join(placeholder_list), image_list
```

*Mantis-Idefics2.* The pre-processing for Mantis-Idefics2 is the same as Idefics2.

*Mantis-BakLLaVA.* The pre-processing for Mantis is as follows:

```
def mantis_preprocess(interleaved_list):
    placeholder_list, image_list = [], []
    for itm in interleaved_list:
        if isinstance(itm, str):
            placeholder_list.append(itm)
        else:
            placeholder_list.append("<image>")
            image_list.append(itm)
    text = "\n".join(placeholder_list)
    # following official guide, use " " to divide image tokens
    text = text.replace("<image>\n", "<image> ")
    return text, image_list
```

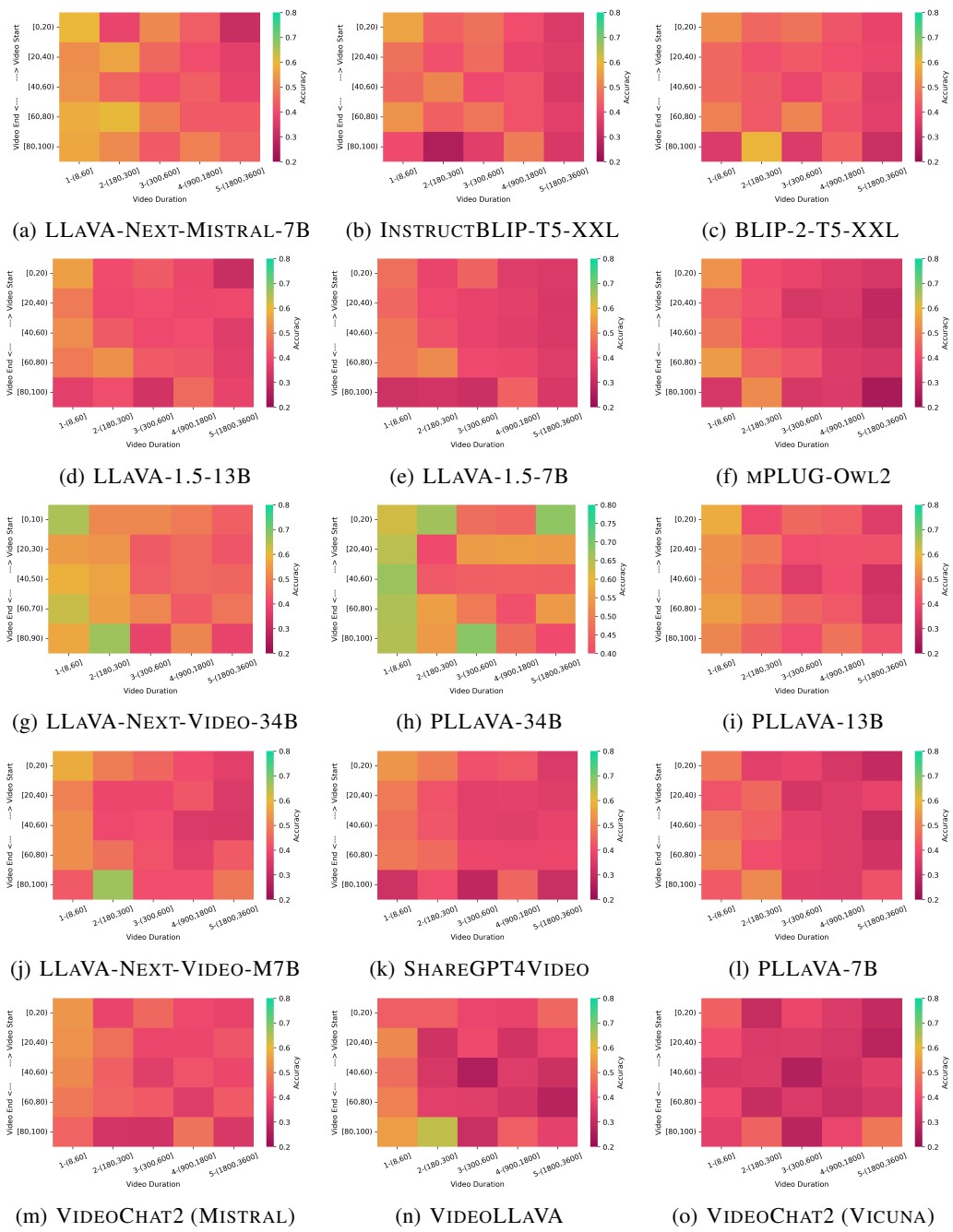

(a) LLaVA-Next-Mistral-7B  (b) InstructBLIP-T5-XXL  (c) BLIP-2-T5-XXL

(d) LLaVA-1.5-13B  (e) LLaVA-1.5-7B  (f) mPLUG-Owl2

(g) LLaVA-Next-Video-34B  (h) PLLaVA-34B  (i) PLLaVA-13B

(j) LLaVA-Next-Video-M7B  (k) ShareGPT4Video  (l) PLLaVA-7B

(m) VideoChat2 (Mistral)  (n) VideoLLaVA  (o) VideoChat2 (Vicuna)

Figure 5: Performance of 15 remaining models *w.r.t.* referring query depth and video duration, as an extension of Fig. 4. Respective analysis in Sec. C.

**Group 2: Image LMMs.** We evaluate all image LMMs [Ye et al., 2023, Dai et al., 2023, Li et al., 2023b, Liu et al., 2023a,b, 2024b] with 8 input frames. As these models can process single images into image embeddings, we concatenate the embeddings from all 8 images along the token dimension and feed the concatenated embeddings into the LMM decoder.

**Group 3: Video LMMs.** We follow the default evaluation settings of all video LMMs with either their official demos or official evaluation scripts. Specifically, PLLaVA, VideoChat2, VideoLLaVA and ShareGPT4Video [Xu et al., 2024, Lin et al., 2023, Chen et al., 2023, 2024] are fed with 16 frames, while LLaVA-Next-Video (34B/M7B) [Zhang et al., 2024] are fed with 32 frames.

All participating models can respond the option letters ("A", "B", "C", "D", "E") to multi-choice questions (MCQ). Thus, we do not include any additional model to assist evaluation.

## C   More Visualizations *w.r.t.* Referring Query Depth

In Fig. 5, we further visualize the performance trend of 14 remaining LMMs with respect the the referring query depth. Most models still roughly follow the general conclusion in Sec. 4.4 that they typically perform worse while the queried moment is closer to *video beginning* or closer to *middle video*, while some models also show no clear trends as their overall performance is limited.

## D   Limitations

Though the LONGVIDEOBENCH has set a meaningful benchmark for multimodal interleaved video-language understanding, in the present stage, its scope only includes vision and language modalities, and not yet included the audio modality. Further, we have not included videos with duration more than 1 hour in our evaluation scope. We will further extend our benchmark into broader scopes.

## E   Broader Impacts

The paper proposes a new evaluation benchmark to advance the development of multimodal AI. Potential societal consequences include the advancement of long-context multimodal large language models and more widespread use of generative AI applications. Despite our effort to remove harmful content, videos in LONGVIDEOBENCHwere sourced from the web and may contain biased, stereotypical information. Therefore, we recommend use of LONGVIDEOBENCH with cautions in research and production.

# F  Details about Human Annotation

## F.1  Instructions and Annotation Interface

**Instructions for (L1) Perception questions.** The specific instructions for each category of (L1) questions are as follows. Each instruction include step-by-step guidance on how a question can be proposed, and provide several examples to further assist understanding.

### I. SCENE-REFERRED EVENT (S2E)

1. Find an action or event.
2. Pause, describe/outline the scene information as the question stem.
3. Use this action or event as the answer.
4. You may refer to these examples:
   - What is the boy in the video doing at Danube Square?
   - What happens after all the ingredients are placed in the pot?
   - When the video transitions to the office, what are the employees doing?
   - What are the characters in the video doing in the café? (non-knowledge videos)
   - What was George Washington doing under the apple tree? (History)
   - What happens in the experiment verifying the surface tension of oil? (Physics)

### II. SCENE-REFERRED OBJECT (S2O)

1. Find a scene, observe the people or objects in this scene.
2. Describe/outline the scene information as the question stem.
3. Use the appearing people/objects and the absent ones as correct and incorrect answers respectively.
4. You may refer to these examples:
   - What objects appeared in Laura's bedroom in the video? (Lifestyle)
   - When all the ingredients are chopped and placed together, which ingredient did not appear? (Cooking)
   - Which communication method was not mentioned in the fourth section? (Physics)
   - Which character did not appear at the duel in the movie?
   - Does the method mentioned later in the video include Y?

### III. SCENE-REFERRED OBJECT ATTRIBUTE (S2A)

1. Find a scene, observe the people or objects in this scene.
2. Describe/outline the scene information and determine an object as the question stem.
3. Use existing and non-existing attributes of the object as correct and incorrect answers respectively, such as material, color, shape, transparency, surface characteristics, structural features.
4. You may refer to these examples:
   - What clothes is Laura wearing in the bedroom with an air conditioner, a bed, and a clothes rack?
   - What color is used to represent the feed forward layer in the Transformer network in Figure 4?
   - Is the person in red clothing wearing glasses in the square with a fountain during the day?
   - What color horse did Napoleon ride in the Battle of Jena?

### IV. EVENT-REFERRED OBJECT (E2O)

1. Find an action or event.

2. Identify the participating people or objects.

3. Describe this action/event as the question stem.

4. Based on the subtitles at the time of the action/event or other background information, detail the participating people/objects as the answer. The options should also be as detailed as possible.

5. You may refer to these examples:
   - Who participated in and won the duel in the movie?
   - Which character finished knitting the sweater?
   - What object exploded in the chemistry experiment in the video?
   - What is the expression of the input variable passed into the Transformer in the video?

## V. OBJECT-REFERRED EVENT (O2E)

1. Find a person or object.

2. Identify the actions/events that happens at their appearance.

3. Describe the person/object as the question stem.

4. Based on a scene where this person/object appears (e.g., first appearance), ask what event happened or what action they took at that time.

5. You may refer to these examples:
   - What did the girl in red do the first time she appeared?
   - What happened the first time a volcano appeared in the video?

## VI. TEXT-REFERRED EVENT (T2E)

1. Find a segment of subtitles, pause the video.

2. Identify the action in the current frame of the video.

3. Think of a few actions that did not appear in the video but are easily confused.

4. Use the action from step 2 as the correct answer, and the actions from step 3 as other options.

5. You may refer to these examples:
   - What was the protagonist doing when mentioning the Renaissance?
   - What event happened when "bidirectional encoder" first appeared in the subtitles?

## VII. TEXT-REFERRED OBJECT (T2O)

1. Find a segment of subtitles, pause the video.

2. Identify a certain object in the frame; for example, a black water bottle.

3. Think of a few objects that did not appear in the video but are easily confused, such as a red water bottle, a black hat, a water dispenser, a transparent water cup.

4. Use the object from step 2 as the correct answer, and the objects from step 3 as other options.

5. You may refer to these examples:
   - What object was present when the lecturer mentioned "revolutionary changes"?
   - Which object did not appear when talking about Jack and Rose having a heart-to-heart conversation?

## VIII. TEXT-REFERRED OBJECT ATTRIBUTE (S2A)

1. Find a segment of subtitles, pause the video.

2. Identify a certain object in the frame.

3. Identify an attribute of the object, such as material, color, shape, transparency, surface characteristics, structural features.

4. Use the object from step 2 as the correct answer, and the attributes from step 3 as other options.

5. You may refer to these examples:
   - What was Tesla's hairstyle like when he was mentioned to have invented alternating current?
   - What color hat was the female protagonist wearing when talking about taking a break?

**Instructions for (L2) Relation questions.** The specific instructions for each category of (L2) questions are as follows. These questions require models to reason over more than one timestamps.

### IX. EVENT BEFORE/AFTER EVENT (E3E)

1. Find two or more adjacent actions or events.
2. Describe one of the actions/events as the question stem, and the other as the correct answer.
3. You may refer to these examples:
   - What did Clara do before taking a photo? (applicable to movie or lifestyle videos)
   - What needs to be done after installing the screws? (applicable to guide videos)
   - Which of the following historical/geographical events was mentioned first? (applicable to history/geography videos)
   - What did the protagonist do before planting the sapling?

### X. OBJECT BEFORE/AFTER OBJECT (O3O)

1. Find two or more people/objects/concepts that appear in the video.
2. Describe one of the objects as the question stem, and the other as the correct answer.
3. You may refer to these examples:
   - After Jack appears, which character appears first in this movie?
   - Which concept is introduced first in the video after entropy is introduced?

### XI. SEQUENCE OF SCENES (SSS)

1. Find multiple scenes (at least three) in the video.
2. Ask questions about the order of these scenes.
3. Answer with the correct sequence and use a few scrambled sequences as distractors.
4. You may refer to this example:
   - Which of the following scene sequences is correct?
   - A. First, a segment of the experiment video is played, then slides with text are shown, and finally XXXX.
   - B. First, slides with text are shown, ...

### XII. SCENE-REFERRED OBJECT TRACKING (SOS)

1. Find a specific person/object/concept that appears in multiple scenes.
2. Define this person/object/concept by their action/attribute in one of the scenes.
3. Then ask in which other scenes did they appear.
4. Distractors are scenes where this object did not appear.
5. You may refer to these examples:
   - In which of the following places did the boy who was running at the beginning of the video appear?
     - A. Square on a sunny day,
     - B. On a boat at sea,
     - C. In a bar on a rainy day, ...
   - In which other scenes did the protagonist's lightsaber, used in the opening fight, appear?

### XIII. SCENE-REFERRED OBJECT ATTRIBUTE CHANGE (SAA)

1. Find a specific person/object/concept that appears in multiple scenes.

2. Define this person/object/concept by their action/attribute in one of the scenes.

3. Then describe another scene and ask what attribute of this person/object/concept has changed at that time.

4. You may refer to these examples:

   - What did the boy running at the beginning of the video change into when climbing the mountain at the end?
     - A. Changed from a white T-shirt to a black vest
     - B. Changed from red shoes to white shoes
     - C. ...
   - What changed in the color of the onions initially poured into the pot?
   - What new part did the sapling planted in the soil at the beginning grow in the later part of the video?

## XIV. EVENT BEFORE/AFTER TEXT (T3E)

1. Find a segment of subtitles, and an action/event in the video that happens before/after it.

2. Rephrase/outline the subtitle as the given information and design the question stem, with the action/event as the correct answer.

3. Distractors are other actions/events in the video that do not meet the sequence relationship in the question stem.

4. You may refer to these examples:

   - What did Clara do after she said, "I eat an apple every day"?
   - What happened before the narrator mentioned the experiment starting?
   - What action was performed after the chef said, "Now wait until the steak surface turns golden"?

## XV. OBJECT BEFORE/AFTER TEXT (T3O)

1. Find the scene where a specific person/object first appears.

2. Then find subtitles before or after this timeframe, rephrase/outline the subtitle as the given information and design the question stem, with the object/person as the correct answer.

3. Distractors are other people/objects in the video that do not meet the sequence relationship in the question stem.

4. You may refer to these examples:

   - Which characters appeared after the commentary mentioned "100 years later"?
   - Which animal appeared on screen before mentioning "dietary habits of North American squirrels"?

## XVI. TEXT-REFERRED OBJECT TRACKING (TOS)

1. Find a specific person/object/concept that appeared at least once along with subtitles.

2. Define this person/object/concept by their action/attribute in one of the scenes.

3. Ask on a subtitle at the object's appearance.

4. Distractors are subtitles where this object did not appear at the corresponding moment.

5. You may refer to these examples:

   - With which subtitles did the boy running at the beginning of the video appear?
   - During which of the following dialogues did the protagonist's lightsaber, used in the opening fight, appear on screen?

## XVII. TEXT-REFERRED OBJECT ATTRIBUTE CHANGE (TAA)

1. Find a specific person/object/concept that appeared at least once along with subtitles.

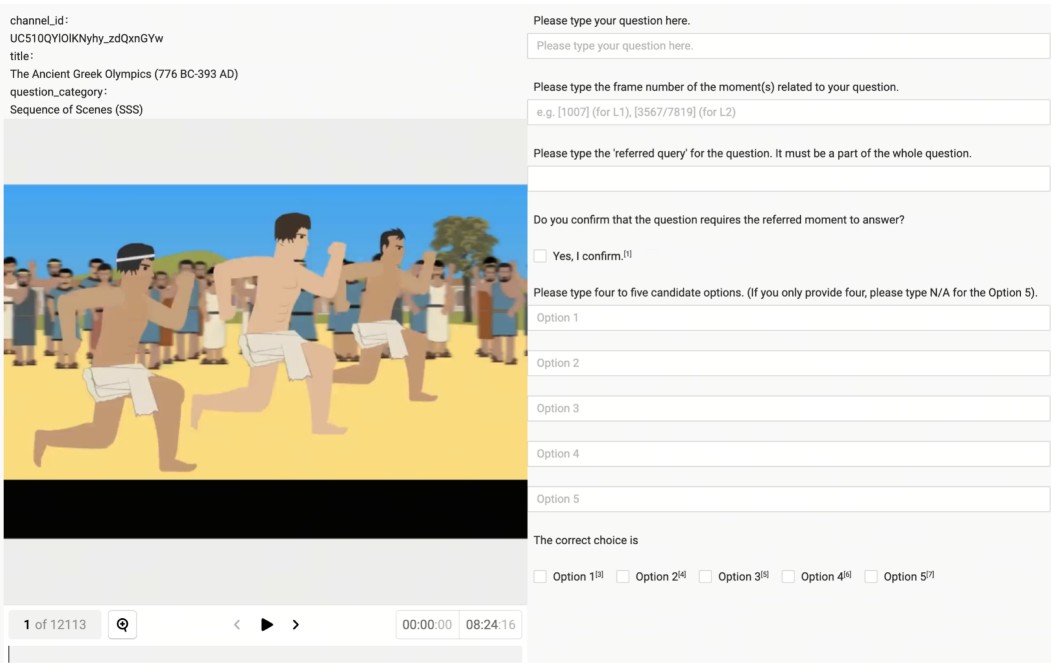

Figure 6: The annotation interface for LONGVIDEOBENCH.

2. Define this person/object/concept by their action/attribute in one of the scenes.

3. Ask what attribute has changed when XX text is mentioned.

4. You may refer to these examples:

   • What change occurred to the girl in the blue jacket and black hood in the middle of the video when mentioning "I am going to sleep"?
       – A. She changed the color of her hood
       – B. She changed into a black jacket
       – C. She took off her hood
       – D. She took off her jacket

**Annotation Interface.** The annotation interface of LONGVIDEOBENCH is illustrated as in Fig. 6. Primarily, a question, 4-5 candidate options, and one correct choice are annotated. Moreover, we explicitly require annotators to highlight the referred query in the question, and ask annotators to label the referred moment. To facilitate annotators to label the moment, the progress bar at left not only shows the current timestamp, but also shows the current frame number (*1 of 12113*).

### F.2 Hourly Wage and Total Compensation

We recruit experienced annotators to perform our annotation. The daily wage for each annotator is 90 USD, with 3 working hours each day, which equals to 30 USD per hour. The total annotation takes 1500 hours (including examination and revision), and the total compensation is 45,000 USD.

### F.3 IRB Approval for Human Study

Please refer to the following dataset sheet for more explanation.

## G    URLs to Websites

Our key URLs are as follows:

Homepage: `https://longvideobench.github.io`
Dataset (Hugging Face): `https://huggingface.co/longvideobench/LongVideoBench`
GitHub Repository: `https://www.github.com/longvideobench/LongVideoBench`

## H Author Statement on Responsibility

```
As the author(s) of the LongVideoBench dataset, we acknowledge
and bear full responsibility for any potential violations of rights,
including but not limited to copyright and privacy issues,
arising from the collection and use of the dataset.

We confirm that the LongVideoBench dataset, which consists of
long videos collected from various web platforms and annotated
by human annotators, will be shared under the Creative Commons
Attribution-NonCommercial-ShareAlike 4.0 (CC BY-NC-SA 4.0) license.
```

## I Hosting, Licensing and Maintenance Plan

We will be hosting the dataset on Hugging Face datasets (`longvideobench/LongVideoBench`). The dataset follows CC-BY-NC-SA-4.0 license, which prohibits any commercial use. We will regularly maintain this dataset and update its instances, with a frequency not lower than twice a year.

## J Dataset Sheet for LONGVIDEOBENCH

| Motivation |
| --- |

**For what purpose was the dataset created?** Was there a specific task in mind? Was there a specific gap that needed to be filled? Please provide a description.

This dataset, referred as LONGVIDEOBENCH, is created to gauge large multimodal models (LMMs) on long-context multimodal understanding, specifically, on long videos. The proposed dataset fills the gap that existing video question-answering usually do not require long frame inputs to answer, thus limited on evaluating long video understanding ability. Our experiments demonstrate that LONGVIDEOBENCH solves this gap and requires hundreds of input frames for most advanced models to reach optimal accuracy.

**Who created this dataset (e.g., which team, research group) and on behalf of which entity (e.g., company, institution, organization)?**

This dataset was created by Haoning Wu, Dongxu Li, Bei Chen, and Junnan Li.

**Who funded the creation of the dataset?** If there is an associated grant, please provide the name of the grantor and the grant name and number.

This work is funded by Rhymes AI.

**Any other comments?**

None.

| Composition |
| --- |

**What do the instances that comprise the dataset represent (e.g., documents, photos, people, countries)?** Are there multiple types of instances (e.g., movies, users, and ratings; people and interactions between them; nodes and edges)? Please provide a description.

The instances include web-sourced videos, and question-answering pairs on these videos. Each question-answering pair include one question, one correct option, and 3-4 distracting options.

**How many instances are there in total (of each type, if appropriate)?**

The LONGVIDEOBENCH includes in-total 3,763 videos, and 6,678 question-answering pairs.

**Does the dataset contain all possible instances or is it a sample (not necessarily random) of instances from a larger set?** If the dataset is a sample, then what is the larger set? Is the sample representative of the larger set (e.g., geographic coverage)? If so, please describe how this representativeness was validated/verified. If it is not representative of the larger set, please describe why not (e.g., to cover a more diverse range of instances, because instances were withheld or unavailable).

The videos are a sample of a larger set from all videos downloaded from 119 web channels, among 10 content categories. We randomly sample uniform population from the 10 content categories. This sampling process is described in main paper Table 3.

**What data does each instance consist of? "Raw" data (e.g., unprocessed text or images) or features?** In either case, please provide a description.

Each video instance consists of "raw" data, directly obtained from web without any feature extraction. Each question-answering instance is from its original annotation.

**Is there a label or target associated with each instance?** If so, please provide a description.

Yes. Each question-answering pair in LONGVIDEOBENCH is associated with a correct option.

**Is any information missing from individual instances?** If so, please provide a description, explaining why this information is missing (e.g., because it was unavailable). This does not include intentionally removed information, but might include, e.g., redacted text.

None.

**Are relationships between individual instances made explicit (e.g., users' movie ratings, social network links)?** If so, please describe how these relationships are made explicit.

No.

**Are there recommended data splits (e.g., training, development/validation, testing)?** If so, please provide a description of these splits, explaining the rationale behind them.

Yes. In LONGVIDEOBENCH, as a benchmark dataset, we don't provide the training set, only *validation* and *test* subsets. The validation set comprises 20% of all data, of which the labels will be released to public for models to decide how to infer on these videos (as we have exemplified in Tables 5/6), and the test set comprises rest 80% of data, of which the labels will be kept hidden, so as to avoid models to overfit on LONGVIDEOBENCH.

**Are there any errors, sources of noise, or redundancies in the dataset?** If so, please provide a description.

No. Our annotation process include error checking and cleaning stages. Please see main paper Section 3.3 for more information.

**Is the dataset self-contained, or does it link to or otherwise rely on external resources (e.g., websites, tweets, other datasets)?** If it links to or relies on external resources, a) are there guarantees that they will exist, and remain constant, over time; b) are there official archival versions of the complete dataset (i.e., including the external resources as they existed at the time the dataset was created); c) are there any restrictions (e.g., licenses, fees) associated with any of the external resources that might apply to a future user? Please provide descriptions of all external resources and any restrictions associated with them, as well as links or other access points, as appropriate.

It is self-contained.

**Does the dataset contain data that might be considered confidential (e.g., data that is protected by legal privilege or by doctor-patient confidentiality, data that includes the content of individuals non-public communications)?** If so, please provide a description.

No.

**Does the dataset contain data that, if viewed directly, might be offensive, insulting, threatening, or might otherwise cause anxiety?** If so, please describe why.

No.

**Does the dataset relate to people?** If not, you may skip the remaining questions in this section.

Yes, LONGVIDEOBENCH is annotated by experienced human annotators.

**Does the dataset identify any subpopulations (e.g., by age, gender)?** If so, please describe how these subpopulations are identified and provide a description of their respective distributions within the dataset.

No.

**Is it possible to identify individuals (i.e., one or more natural persons), either directly or indirectly (i.e., in combination with other data) from the dataset?** If so, please describe how.

Yes. Some videos include text subtitles that identify names of individuals, *e.g.* movie characters.

**Does the dataset contain data that might be considered sensitive in any way (e.g., data that reveals racial or ethnic origins, sexual orientations, religious beliefs, political opinions or union memberships, or locations; financial or health data; biometric or genetic data; forms of government identification, such as social security numbers; criminal history)?** If so, please provide a description.

No.

**Any other comments?**

None.

---

| **Collection Process** |
|---|

**How was the data associated with each instance acquired?** Was the data directly observable (e.g., raw text, movie ratings), reported by subjects (e.g., survey responses), or indirectly inferred/derived from other data (e.g., part-of-speech tags, model-based guesses for age or language)? If data was reported by subjects or indirectly inferred/derived from other data, was the data validated/verified? If so, please describe how.

The videos are raw videos that are directly observable. The question-answering pairs are annotated by human, and verified one by one. Specifically, we employ an examiner to check the annotation correctness, and a reviser to revise the annotations labeled as incorrect by examiners.

**What mechanisms or procedures were used to collect the data (e.g., hardware apparatus or sensor, manual human curation, software program, software API)?** How were these mechanisms or procedures validated?

We use official software APIs to download videos from platforms.

**If the dataset is a sample from a larger set, what was the sampling strategy (e.g., deterministic, probabilistic with specific sampling probabilities)?**

The sampling is a uniform random sampling.

**Who was involved in the data collection process (e.g., students, crowdworkers, contractors) and how were they compensated (e.g., how much were crowdworkers paid)?**

Contractors. They were paid 90 USD per day, with 3 working hours each day to avoid fatigue.

**Over what timeframe was the data collected? Does this timeframe match the creation timeframe of the data associated with the instances (e.g., recent crawl of old news articles)?** If not, please describe the timeframe in which the data associated with the instances was created.

The videos were collected during April 2024. The creation timeframe of the data associated with the instances are no later than April 2024, but may be as early as 2010 (while the videos are uploaded to web platforms).

**Were any ethical review processes conducted (e.g., by an institutional review board)?** If so, please provide a description of these review processes, including the outcomes, as well as a link or other access point to any supporting documentation.

Yes. It will be annouced in later versions.

**Does the dataset relate to people?** If not, you may skip the remaining questions in this section.

Yes.

**Did you collect the data from the individuals in question directly, or obtain it via third parties or other sources (e.g., websites)?**

Yes, we collect the data from the annotators in question directly.

**Were the individuals in question notified about the data collection?** If so, please describe (or show with screenshots or other information) how notice was provided, and provide a link or other access point to, or otherwise reproduce, the exact language of the notification itself.

Yes, prior to the experiment, participants are provided with a brief outlining the data collection process, and they must sign it to indicate their agreement. The brief is as follows:

```
You will participate to annotate questions and answers on videos.
Your duties include as follows:

1. Participate in our mandatory training to understand the guidelines
of annotation.

2. Watch videos, and provide annotations on these videos.

Each annotation includes the following terms:

(a) A question;
(b) One or more timestamp(s) on the question;
(c) Four to five options;
(d) A checkbox to pick the correct option.

3. Check the correctness of annotations from other annotators.

4. Report videos that are not appropriate during the process.
```

**Did the individuals in question consent to the collection and use of their data?** If so, please describe (or show with screenshots or other information) how consent was requested and provided, and provide a link or other access point to, or otherwise reproduce, the exact language to which the individuals consented.

Yes. After reading the document, each annotator needs to sign the following form:

```
    I aware the duties of the annotation process.
```

```
I understand that the annotated data will be published.
I will not put any personal information into annotations.

Signed by: [ANNOTATOR's NAME]
```

**If consent was obtained, were the consenting individuals provided with a mechanism to revoke their consent in the future or for certain uses?** If so, please provide a description, as well as a link or other access point to the mechanism (if appropriate).

Yes, the participants are allowed to quit the experiment and revoke their consent at any time of the experiment. The completed annotated instances before the revoke will be paid.

**Has an analysis of the potential impact of the dataset and its use on data subjects (e.g., a data protection impact analysis) been conducted?** If so, please provide a description of this analysis, including the outcomes, as well as a link or other access point to any supporting documentation.

Yes. We have examined each collected video and made sure that all videos are public-domain videos without containing private or proprietary information from any people; for the annotations, we validate that they do not contain any private information from the annotators. We will continuously maintain the dataset and remove any video from our dataset (and their associating question-answering pairs) if it becomes non-public.

**Any other comments?**

None.

| Preprocessing/cleaning/labeling |
| --- |

**Was any preprocessing/cleaning/labeling of the data done (e.g., discretization or bucketing, tokenization, part-of-speech tagging, SIFT feature extraction, removal of instances, processing of missing values)?** If so, please provide a description. If not, you may skip the remainder of the questions in this section.

The LONGVIDEOBENCH contains human-annotated question-answering pairs to the "raw" videos. For each video, we annotate up to three question-answering pairs, while each question-answering pair contains a question about the video, a correct answer to the question, and several distracting options to it.

**Was the "raw" data saved in addition to the preprocessed/cleaned/labeled data (e.g., to support unanticipated future uses)?** If so, please provide a link or other access point to the "raw" data.

Yes. All raw videos are provided in the dataset, as they are still a main composing part of the dataset.

**Is the software used to preprocess/clean/label the instances available?** If so, please provide a link or other access point.

No. LONGVIDEOBENCH is labeled by human, with our proprietary system which is not intended to be open-sourced.

**Any other comments?**

None.

| Uses |
| --- |

**Has the dataset been used for any tasks already?** If so, please provide a description.

This dataset is used for evaluating large multimodal models (LMMs) on long-context understanding abilities in same paper that proposes it.

**Is there a repository that links to any or all papers or systems that use the dataset?** If so, please provide a link or other access point.

N/A. Only we use the proposed dataset now. We will maintain this in the future.

**What (other) tasks could the dataset be used for?**

N/A.

**Is there anything about the composition of the dataset or the way it was collected and preprocessed/cleaned/labeled that might impact future uses?** For example, is there anything that a future user might need to know to avoid uses that could result in unfair treatment of individuals or groups (e.g., stereotyping, quality of service issues) or other undesirable harms (e.g., financial harms, legal risks) If so, please provide a description. Is there anything a future user could do to mitigate these undesirable harms?

N/A.

**Are there tasks for which the dataset should not be used?** If so, please provide a description.

The dataset should not, in any circumstances, be used for commercial applications. Any further post-processing or sub-sampling to demonstrate any bias (e.g. racial, gender) is strictly prohibited.

**Any other comments?**

None.

| Distribution |
|:---:|

**Will the dataset be distributed to third parties outside of the entity (e.g., company, institution, organization) on behalf of which the dataset was created?** If so, please provide a description.

No.

**How will the dataset will be distributed (e.g., tarball on website, API, GitHub)** Does the dataset have a digital object identifier (DOI)?

It has been distributed on Hugging Face datasets. It does not have a DOI.

**When will the dataset be distributed?**

The dataset has already been released.

**Will the dataset be distributed under a copyright or other intellectual property (IP) license, and/or under applicable terms of use (ToU)?** If so, please describe this license and/or ToU, and provide a link or other access point to, or otherwise reproduce, any relevant licensing terms or ToU, as well as any fees associated with these restrictions.

This dataset will be distributed under Creative Commons Attribution Non Commercial Share Alike 4.0 (CC-BY-NC-SA 4.0) license. No commercial use is allowed.

**Have any third parties imposed IP-based or other restrictions on the data associated with the instances?** If so, please describe these restrictions, and provide a link or other access point to, or otherwise reproduce, any relevant licensing terms, as well as any fees associated with these restrictions.

No.

**Do any export controls or other regulatory restrictions apply to the dataset or to individual instances?** If so, please describe these restrictions, and provide a link or other access point to, or otherwise reproduce, any supporting documentation.

No.

**Any other comments?**

None.

| Maintenance |
| :---: |

**Who will be supporting/hosting/maintaining the dataset?**

The authors will be maintaining the dataset.

**How can the owner/curator/manager of the dataset be contacted (e.g., email address)?**

Please contact primary owner Haoning Wu, at `realtimothyhwu@gmail.com` .

**Is there an erratum?** If so, please provide a link or other access point.

No.

**Will the dataset be updated (e.g., to correct labeling errors, add new instances, delete instances)?** If so, please describe how often, by whom, and how updates will be communicated to users (e.g., mailing list, GitHub)?

Yes. We will review the dataset twice a year and updates will be noticed on Github and Hugging Face homepages.

**If the dataset relates to people, are there applicable limits on the retention of the data associated with the instances (e.g., were individuals in question told that their data would be retained for a fixed period of time and then deleted)?** If so, please describe these limits and explain how they will be enforced.

N/A. The annotators are aware that their annotations will be public for unlimited time period.

**Will older versions of the dataset continue to be supported/hosted/maintained?** If so, please describe how. If not, please describe how its obsolescence will be communicated to users.

Yes. The older versions of the dataset have independent links for download.

**If others want to extend/augment/build on/contribute to the dataset, is there a mechanism for them to do so?** If so, please provide a description. Will these contributions be validated/verified? If so, please describe how. If not, why not? Is there a process for communicating/distributing these contributions to other users? If so, please provide a description.

Yes, they can contact the owner and their contributions will be reviewed by the owner.

**Any other comments?**

None.

