# OpenReview forum: "LongVideoBench: A Benchmark for Long-context Interleaved Video-Language Understanding"
_NeurIPS.cc/2024/Datasets_and_Benchmarks_Track — NeurIPS 2024 Track Datasets and Benchmarks Poster_

### Official Review · Reviewer_ZDwx · 2024-07-22
**LongVideoBench: A Benchmark for Long-context Interleaved Video-Language Understanding**

**Rating:** 5
**Confidence:** 5
**Correctness:** Yes.
**Clarity:** Overall, this paper is well written a…

**Review:**

Pros:
The paper is well-written and clear.
The paper provides a thorough evaluation of both proprietary and open-source models, focusing on their ability to handle long-context multimodal modeling.
Cons:
Lack of long temporal dynamic context. Most of the questions involve static objects or scene changes, rather than the temporal dynamic behaviors unique to video. The example in Figure 1 can be solved using a single-frame vision-language model. Therefore, I think the design of these questions does not reflect the essence of long video understanding.
Like EgoSchema, they introduce the Certificate Length to reflect the length in long-term understanding. However, this paperr also annotates videos of different lengths, it does not reflect the concept of length.

**Strengths:**

- The paper presents a new benchmark for video-language interleaved inputs, addressing a gap in existing benchmarks for long video understanding.
- The benchmark covers a wide range of video lengths and themes, with diverse question types.
- A new referring reasoning task is proposed to evaluate the abilities of LMMS.

**Additional Feedback:**

N/A

**Documentation:**

Yes, I believe so.

**Limitations:**

The authors indeed acknowledge the limitations of the modality and video length. However, it is more important to consider what are the essential problems that long videos need to solve.

**Opportunities For Improvement:**

- More temporal dynamic context
- Evaluate the Certificate Length

**Relation To Prior Work:**

Yes

**Summary And Contributions:**

This paper introduces LONGVIDEOBENCH, a novel benchmark designed to evaluate Large Multimodal Models (LMMs) in understanding video and subtitle inputs up to an hour long. The benchmark includes 3,763 web-collected videos across diverse themes, with 6,678 multiple-choice questions in 17 fine-grained categories. Evaluations indicate that even the most advanced proprietary models (e.g., GPT-4o, Gemini-1.5-Pro) face significant challenges, and open-source models show an even larger performance gap. The results also suggest that model performance improves only when they process more frames, positioning LONGVIDEOBENCH as a valuable benchmark for evaluating future-generation LMMs.

---

> ### Author Rebuttal · Authors · 2024-08-16
>
> Thanks for confirming the quality of writing, documentation, and experiments. Please find our response below.
>
> **Q1:** How does the design of LongVideoBench pose challenges to long-term video modeling?
>
> **A1:** The main principle for benchmark design, that reflects challenges in long video modeling, is the referring reasoning questions.
>
> To make an analogy, the "referring" process is related to the Needle-In-A-Haystack (NIAH) task, which is well-recognized for evaluating LLM long context modeling. The "reasoning" process adds an extra layer of challenges. Combined they require advanced multimodal long-context modeling and reasoning capabilities.
>
> Retrieving and reasoning about details from long videos is non-trivial. As suggested by Table 7, the best open-source model (IDEFICS2) can solve less than half questions (44.7%) for the longest video group [900,3600]. This shows challenges present by our benchmark for long video understanding.
>
> **Q2:** Clarification on confusion: "Questions can be solved using single-frame vision-language models."
>
> **A2:** As suggested in Table 5, with 256 frames as opposed to a single frame as input,
> - GPT-4o succeeds on 66.7% v.s. 41.7% (+60% relative, +25% absolute)
> - Gemini-1.5-Pro succeeds on 64.7% v.s. 38.6% (+68% relative, +26.1% absolute)
> This shows that a significant portion of questions in LongVideoBench cannot be solved using a single frame model.
>
> Additionally, we would also like to clarify that the example in Figure 1 cannot be solved using a single frame model either. [The video](https://github.com/longvideobench/longvideobench.github.io/raw/main/static/videos/banner_video.mp4) contains lots of people while more than one has jacket or backpack. Therefore, to correctly answer this question, it requires reasoning over multiple moments from a durational video in order to correctly locate the particular referred person (*a woman with headband tied to her head*), and the particular referred scene (*when she comes down the hill*), instead of being solved using single-frame models.
>
> **Q3:** Clarification on confusion "most of the questions involve static objects".
>
> **A3:** LongVideoBench contains 17 types of questions (Table 2 for definition, Figure 2 for examples), covering extensive questions on event progression, attribute changes, object tracking and attribute tracking.
>
> Among them, questions below are directly about dynamic video context.
>
> - 6 question types are directly related to actions, movements, see Scene-referred Event (S2E), Object-referred Event (O2E), Event before/after Event (E3E), Event before/after Text (T3E) in Figure 2, for example.
> - 4 question types are directly related to object tracking / attribute change along the video: Scene-referred Object Tracking (SOS), Scene-referred Object Attribute Change (SAA), Text-referred Object Tracking (TOS), Text-referred Object Attribute Change (TAA) in Figure 2, for example.
>
> In addition, due to the design principle of referring reasoning, even questions related to objects and scenes require models' capabilities for processing long input videos.
>
> **Summary:** LongVideoBench features a large portion of questions on dynamic video context (**A3**), underpinned by the principal design of referring reasoning questions (**A1**). It challenges directly and significantly the long-video understanding capabilities of multimodal models. We are willing to answer more questions and would appreciate the reviewer raising the score.

---

> > ### Comment · Area_Chair_b8h5 · 2024-08-29
> > **Response to author rebuttal**
> >
> > Dear Reviewer,
> >
> > The ddl for author and reviewer discussion is approaching. Please check the author rebuttal and leave some comments to respond to author rebuttal.
> >
> > Thanks,
> >
> > Your AC

---

> > ### Author Response · Authors · 2024-08-30
> > **Have we addressed your questions?**
> >
> > Hi reviewer, we'd like to know whether your questions have now be clarified? If your questions are addressed, please reconsider your review. Otherwise, we'd be happy to further the discussion.
> >
> > Thanks.

---

### Official Review · Reviewer_fQc5 · 2024-07-25
**Fairly good paper but lack of novelty**

**Rating:** 5
**Confidence:** 5
**Correctness:** The claims are correct.
**Clarity:** The paper is well-written and easy to…

**Review:**

Overall, the paper is well-written and easy-to-follow. The motivation of the proposed benchmark is reasonable -- some existing benchmarks  for video understanding are focusing on seconds-long videos, and more importantly, they suffer from severe single-frame biases. In this work, the authors tend to address these issues by annotating a new video understanding benchmark specifically for long videos. However, the proposed benchmark has some drawbacks such as lack of novelty that make the proposed work stand out from tons of long video benchmarks. Please see the detailed pros and cons in the following sections.

**Strengths:**

1. The motivation of the proposed benchmark is clear. That is, introduce a benchmark for long (up to one hour) video understanding and try to address the single-frame bias problem.
2. The annotations shall be of high quality, as they are manually annotated.
3. The authors provide extensive evaluation of existing Video-LLMs on the proposed LongVideoBench.

**Additional Feedback:**

Overall, I have not much concern about the data annotation & organization process. My major concern is that the proposed benchmark is somewhat too similar to existing ones that either 1) containing both short and long videos, and 2) specially designed for long video understanding. It would be better if the authors could clarify the significance of the proposed LongVideoBench.

**Documentation:**

The details about data collection and organization are sufficient.

**Limitations:**

The authors have addressed the limitations and potential negative societal impact of their work.

**Opportunities For Improvement:**

1. The novelty of this work seems limited. Evaluating Video-LLMs on longer videos (from minutes to hours) has been a clear and well-studied area in the past months. And a lot of works have addressed this problem by introducing long video benchmarks. I do not see any characteristics of the proposed work that make it significantly different from its counterparts.
2. It would be better if the authors could provide a solution for long video understanding after benchmarking and analyzing the results.

**Relation To Prior Work:**

No. The proposed benchmark seems like yet another video understanding benchmark trying to solve long video problems. I do not see any significant differences between the proposed benchmark and existing ones.

**Summary And Contributions:**

This paper introduces LongVideoBench, a comprehensive benchmark for long video understanding. The proposed benchmark differs from existing ones in terms of two aspects: 1) the lengths of videos in LongVideoBench are much longer than that in existing benchmarks, and 2) the authors proposed referring reasoning to address the problem of single-frame bias. Extensive evaluations of existing Video-LLMs are conducted on the proposed benchmark, and they reveal the significant challenges in long video understanding.

---

> ### Author Rebuttal · Authors · 2024-08-16
>
> We thank the reviewer for recognizing LongVideoBench as a good paper with good motivation, high-quality annotation, and extensive evaluation. To avoid misunderstanding on novelty-related concerns, we carefully respond to these concerns as follows:
>
> **Q1:** Difference with prior work / Clarification on misunderstanding "yet another video understanding benchmark"
>
> **A1:** It is noteworthy that LongVideoBench is the first public video benchmark to show a consistent gain with growing input frames up to 256, while most prior benchmarks saturate quickly with 16-64 frames. This is important to measure the increasing long-context window size of multimodal models.
> For example, on LongVideoBench, GPT-4o grows consistently from 41.7 (1 frame) --> 58 (16 frames) --> 62 (64 frames) --> 66.7 (256 frames).
>
> Below, we compare with the most alike benchmarks as we are aware of for long video understanding.
> - **1H-VideoQA** (Feb 2024, Google in-house evaluation benchmark). Gemini shows consistent performance gain on 1H-Video up to 1fps. Yet, the benchmark is not public. No other open-source models' performance is reported or analyzed.
> - **EgoSchema** (Aug 2023, Stanford) contains 3-min single-scene egocentric-only videos. On Egoschema, Gemini-1.5 Pro solves 53.6% questions using a mere single frame, while improving merely from 70.2% (16 frames) to 72.7% (150 frames), with a 2.5% absolute increase. see https://arxiv.org/pdf/2403.05530 page 24.
> In contrast, on LongVideoBench Gemini-1.5 Pro improves steadily with more frames from 52.7% (16-frames) --> 58.6% (64 frames) --> 61.9% (128 frames) --> 64% (256 frames).
>
> This demonstrates that LongVideoBench is advantageous for measuring progress of LMMs of longer context windows.
> We respectfully request the reviewer to nominate the mentioned work, so that we can further clarify differences and highlight the significance of LongVideoBench.
>
> **Q2:** "Evaluating Video-LLMs on longer videos (from minutes to hours) has been a clear and well-studied area in the past months", or is it?
>
> **A2:** We are not confident that long video understanding "has been well-studied". On LongVideoBench, even the best proprietary LMMs fail a significant portion of LongVideoBench questions (33-41%), and the open-source counterparts fail in 51-64% cases. In addition, most open-source models cannot effectively take over 32 frames, and even drop to almost as random guess with 64 frames or more.
>
> Such new discoveries revealed by LongVideoBench are significant in themselves and unveil vast research opportunities for developing long context window LMMs. These findings are highly non-trivial and are not possible without the design of referring reasoning questions or the careful data annotation and evaluation efforts.
>
> **Summary:** LongVideoBench is the first long video understanding benchmark proven effective for increasing context windows for open research (**A1**). With careful benchmark design and development, evaluation results shed light on the significant challenges ahead for developing long-context LMMs (**A2**).
>
> We appreciate the reviewer reconsider their decision and raise the score.

---

> > ### Comment · Area_Chair_b8h5 · 2024-08-29
> > **Response to author rebuttal**
> >
> > Dear Reviewer,
> >
> > The ddl for author and reviewer discussion is approaching. Please check the author rebuttal and leave some comments to respond to author rebuttal.
> >
> > Thanks,
> >
> > Your AC

---

> > ### Author Response · Authors · 2024-08-30
> > **Have we addressed your questions?**
> >
> > Hi reviewer, we'd like to know whether your questions have now be clarified? If your questions are addressed, please reconsider your review. Otherwise, we'd be happy to further the discussion.
> >
> > Thanks.

---

> > > ### Comment · Reviewer_fQc5 · 2024-08-31
> > >
> > > Thanks for the response from the authors. However, my concerns regarding the novelty of this work are not fully addressed. I do not see the significance of the proposed work compared with many existing counterparts, e.g., EgoSchema, MLVU, Video-MME, thus I'm holding my original rating.

---

> > > > ### Author Rebuttal · Authors · 2024-08-31
> > > >
> > > > The significance of our benchmark compared to EgoSchema has been extensively proved in our paper and further highlighted in our response. We respectfully ask the reviewer to provide justifications on the reasons for disagreement, despite us having no time to reply.
> > > >
> > > >
> > > > As for MLVU and Video-MME, both papers appeared on arXiv after the NeurIPS abstract submission deadline (May 29th). MLVU appeared online on May 31th while Video-MME on June 6th. According to the official policy of NeurIPS 2024: "papers that appeared online within two months of a submission will generally be considered 'contemporaneous' in the sense that the submission **will not be rejected** on the basis of the comparison to contemporaneous work." Therefore, we would like to remind the reviewer that it is **invalid** to recommend rejection based on such groundings.
> > > >
> > > >
> > > > Although irrelevant to our paper’s rating, we are more than happy to discuss the major difference between our benchmark and the aforementioned MLVU and Video-MME benchmarks: we design a novel referring reasoning task that systematically demonstrates various models’ performance gain by progressively increasing the number of input frames. Therefore, our LongVideoBench is advantageous for measuring progress of LMMs in understanding longer videos.

---

### Official Review · Reviewer_S89L · 2024-07-25
**Carefully curated benchmark for long-context Video Understanding with in-depth analysis over video context.**

**Rating:** 8
**Confidence:** 4

**Review:**

Different from prior video-language benchmark, the benchmark handles long context video understanding with the help of referring query to help the models to narrow down the specific moments in the video. The analysis covers diverse number of input frames and validates that the benchmark requires large number of frames to perform well in their long context video.  The dataset is also carefully curated with three rounds of annotation stages and maintains its high quality.

**Pros**
- Dataset is of high quality and carefully curated with another round of validation.
- Collects and investigates varying length of videos from 10 seconds to 1 hour, which is aligned with the original motivation of long-context interleaved video understanding. This is different from prior work that deals either with just trimmed videos or long videos.
    - Appropriately addresses related long video benchmark such as EgoSchema which still can be solved with limited number of frames, unlike this dataset.
- Experiments are performed for a diverse set of models from propreitary to open long-context, image-based, and video LMMs.
- Multimodal needle in haystack problem is observed in LLMs with the help of FIgure 4 that effectively demonstrates the problem of identifying moments in the middle of the video.

**Cons**
- Missing LLM only baseline to understand the potential text bias in the reasoning questions.
   - A sophisticated prompting strategy, such as integrating dense captions and socratic based model, or chain of thought would be interesting to see as well.
- Missing human performance to determine the upper bound performance on the dataset.
- Missing qualitative examples of model performance and failure case to understand the performance bottleneck of the models.

**Strengths:**

Dataset motivation and experiment results are well aligned to test the LMM performance on varying video context length, and authors appropriately investigate the needle in haystack problem in long context video understanding.  Dataset is carefully curated and manually annotated by humans to guarantee high quality with appropriate scale.
See also the pros above.

**Additional Feedback:**

None

**Clarity:**

paper is well written with qualitative examples showing the overview of the dataset.

**Correctness:**

Dataset is validated thoroughly and evaluation methods are appropriately proposed as multiple choice format.

**Documentation:**

Yes the data collection and organization are well documented in the appendix.

**Limitations:**

Limitation is addressed as there is more room for improvement to deal with long context videos.

**Opportunities For Improvement:**

- Add qualitative examples and failure cases for readers to understand the limitations of exiting LLMs.
- Efficient prompting strategies with socratic methods could be considered similar to [A].
- See also the cons above.


[A]: A Simple LLM Framework for Long-Range Video Question-Answering

**Relation To Prior Work:**

Compares with EgoSchema as alternative long-duration video understanding benchmark in related work. The authors address that EgoSchema often limit their scope to videos on specific themes and still include a large proportion of summary questions solvable with limited frames. The introduced LongVideoBench by design requires dense input frames to solve, encompassing diverse video topics and varying lengths up to hour long with the help of referring query.

**Summary And Contributions:**

LongVideoBench is a VideoQA benchmark that assesses large multimodal models (LMMs) on their ability to understand videos of varying lengths, from 10 seconds to an hour. This benchmark differs from previous ones by including localized information about specific moments in the videos. It uses a referring query that describes a moment from the video followed by a question that requires the models to reason over the referred context to deduce the answer. The questions are categorized into perception and relation levels, encompassing 17 types of referring reasoning questions. The dataset undergoes a three-stage process: initial annotation, validation of question categorization, and revision of incorrectly labeled annotations. Subtitles are also collected to enhance the LMMs' ability to leverage video-text information.

In their experiments, the authors evaluate 20 LMMs across four categories: 1) Proprietary LMMs (GPT4-o & Gemini), 2) Open-source long context LMMs, 3) Image context LMMs, and 4) Video LMMs. They first analyze the models' performance across different video lengths and a range of frames from 1 to 256. Proprietary models show improved performance with an increased number of frames, particularly in longer video contexts, although performance generally decreases with video length due to the multimodal haystack problem. GPT4-o and Gemini effectively utilize subtitles to boost their performance, whereas other open-source models struggle with the integrated video-text information. Final results indicate that proprietary models significantly outperform open-source models (66.7% vs 53.2%). Surprisingly, 7B video LMMs do not surpass the performance of image-based 7B LMMs. However, increasing the size of the LLMs markedly improves performance (from 45.6% to 53.2% moving from pLLAVA 7B to 32B). The authors also detail performance variations by query depth and video length in Figure 4, highlighting the needle-in-a-haystack problem where LLMs perform worse on questions located in the middle of the video compared to those at the beginning or end, due to the challenge of maintaining long-term video context.

---

> ### Author Rebuttal · Authors · 2024-08-16
>
> Thanks for recognizing the careful curation and analysis. We are encouraged to see that the key motivation of LongVideoBench is recognized, including our endeavor to address the few-frame bias in video benchmarks, and needle-in-a-haystack (NIAH) experiments to diagnose LLMs and LMMs on long-context understanding.
>
> We also appreciate the constructive suggestions. Please see our response as follows.
>
> **Q1:** 'LLM-only baseline to understand potential text bias in reasoning questions.'
>
> **A1:** Thanks for the suggestion.
> - **Multimodal models with text-only inputs**: in Table 6, we evaluate multimodal models with only text inputs (subtitles + questions + candidates). For example, GPT-4o and Gemini-1.5-Pro without any images achived 44.6% and 43.0% accuracy without frames. In comparison, their vision-included results are 66.7% and 64.0% respectively.  For open-source Idefics2-8B, it reaches 25.6% accuracy with only text input (49.7% with vision).
> - **Language-only models.** As suggested by the reivewer, we further experimented with LLaMa-3.1-8B-Instruct and Mistral-v0.3-Instruct-7B, two latest open-source language-only long-context models. On LongVideoBench, they achieve 36.2% and 28.4% accuracy respectively. These results are significantly lower than those with video inputs.
> With these observations, we conclude that there does exist slight text bias in our question-answers. Possible reasons might include unlikely false options or contextual correlation between the positive option with the referred query in our question. We appreciate the suggestion and would include results and analysis into the revision.
>
> **Q2:** More Evaluation Strategies.
>
> **A2:** Thank you for this feedback. As suggested, we further evaluate gpt-4o-2024-05-13, the best existing LLMs at the time of submission, on two extra settings:
>
> (1) **Caption Each Frame, and Answer.**
>
> In this setting, we first obtain from GPT-4o a caption of a reasonable-length (50-words) for each frame.  We append each caption after the corresponding frame. Finally we prompt GPT-4o to reason over the interleaved sequence of frames and their captions. Results show a slight performance drop from 63.5% to 62.9%. Our observations on this result are below:
> 1.. Frame captions fail to capture temporal continuation of scenes, objects and attributes, which are necessary to solve LongVideoBench questions.
> 2.. Frame captions do not provide much complementary information to assist in video understanding. Unlike subtitles, information in captions mostly overlapped with the visuals and are even less faithful due to captioning errors.
> In addition, we further experimented with using these frame-by-frame captions in-lieu-of the frames. GPT-4o reaches 48.7% accuracy on this 'caption-instead-of-frame' setting, better than the text-only setting (44.9%), suggesting that dense captioning contains some but not sufficient information required to answer LongVideoBench, demonstrating the necessity for long-context visual understanding in our benchmark.
>
> (2) **Think Step by Step. (CoT)**
>
> In this setting, we prompted 4o with the instruction "Please think step by step, and finally answer the question." On LongVideoBench validation set with 128 input frames, GPT-4o's performance drops slightly from 63.5% to 63.1%. This result looks counter-intuitive and we have thus further observations below:
> 1. With CoT, GPT-4o tends to be more thoughtful and refused to answer 2.9% more questions due to "insufficient information". This led primarily to the performance drop.
> 2. If not accounting for the question subsets refused to answer, the setting with CoT achieved 1.3% better than the non-CoT version.
> 3. Even with the 1.3% improvement, the CoT capabilities of GPT-4o in understanding long videos are not as impressive. This reveals the significant challenge presented by LongVideoBench and opens up research questions on improving CoT capabilities for long videos.
> We believe these are intriguing results and will include them in the revised manuscript.
>
> **Q3:** About Human Performance.
>
> **A3:** LongVideoBench is labeled by human annotators with perception and reasoning questions that **require no expert knowledge**. Given sufficient time to view the video and retrieve required details, we expect a patient human to answer most questions, with mistakes by accident.
>
> To ensure the labeling quality, we also introduced a examination step following the annotation step, where we have hired examiners to check each annotation. We kept the annotation only if the examiner agreed with the annotator. If not, the question is sent back for revision, or discarded in the cases of insufficient quality of question annotation.
>
> **Q4:** About qualitative examples.
>
> **A4:** Thanks for pointing this out. We agree this is important and will add success and failure cases of different models for qualitative analysis in our revision.

---

> > ### Comment · Area_Chair_b8h5 · 2024-08-29
> > **Response to author rebuttal**
> >
> > Dear Reviewer,
> >
> > The ddl for author and reviewer discussion is approaching. Please check the author rebuttal and leave some comments to respond to author rebuttal.
> >
> > Thanks,
> >
> > Your AC

---

> > ### Author Response · Authors · 2024-08-30
> > **Thanks for the feedback.**
> >
> > We appreciate your valuable feedback and have added experiments results accordingly. Thanks.

---

### Author Response · Authors · 2024-08-30
**Please respond to rebuttal**

Hi Reviewers,

We have put significant efforts in addressing questions in the review and would appreciate your feedback.

- @Reviewer S89L: we appreciate you valuable feedback and have added experiments results accordingly. Thanks.
- @Reviewer fQc5 and ZDwx: since you gave "marginally below acceptance" initial reviews, we'd like to know whether your questions have now be clarified? If your questions are addressed, please reconsider your review. Otherwise, we'd be glad to further the discussion.

@Area Chairs: We appreciate your kind efforts in initiating the discussion. We'd like to re-iterate:

- LongVideoBench is the first public benchmark shows consistent gains by increasing frames up to 256 frames.

-- This is important to assess the long-context reasoning capabilities of LMMs for future development.

-- This is non-trivial and not possible without the careful dataset design and curation - as evidenced by that no prior benchmark has achieved so.

- We believe the benchmark makes sufficient contribution, provides valuable insights into the important task, and opens up vast research opportunities in exploring long-context LMMs, thus making it a strong submission for the dataset and benchmark track.

---

### Decision · Program_Chairs · 2024-09-26

**Decision:**

Accept (Poster)

**Comment:**

This paper receives mixed reviews: one accept and two borderline rejects. This paper has some merits of proposing a new long video benchmark where up to 256 frames can still contribute to a performance imporovement. Two reviewers still raise some concerns on the  proposed benchmark. For example, some similarity to previous long video benchmarks, unclear temporal certificate for answering the question, lack of human performance. The author provides a detailed rebuttal to these concerns.. After a careful check of paper, rebuttal, reviewer comments, the AC thinks the rebuttal has sucessfully addressed the major concern. In particular, the referring QA is a new important task for video understanding. The proposed benchmark has notable difference to previous benchmarks such as MovieChat (CVPR 2024). Compared with the traditional general QA task, the referring reasoning task is more challenging as it requires temporal grounding to first localize the object correctly and is more valuable as it can test fine-grained understanding ability of MLLM. Thus, the AC makes an accept recommendation.

Another suggestion: the AC thinks if the authors aim to propose a new QA type of referring reasoning, the evaluation metric might be not enough with a simple accuracy for QA. It would be better design a new metric related to grounding to check whether the model has identified the correct object to answer the question.